

# Light absorbing particles and snow aging feedback enhances albedo reduction on the Southwest Greenland ice sheet

Isatis M. Cintron-Rodriguez[1], Åsa K. Rennermalm[2], Susan Kaspari[3], Sasha Z. Leidman[2]

[1]Department of Environmental Sciences, Rutgers, The State University of New Jersey, New Brunswick 08901, New Jersey, USA
[2]Department of Geography, Rutgers, The State University of New Jersey, New Brunswick 08901, New Jersey, USA
[3]Department of Geological Sciences, Central Washington University, Ellensburg, WA 98926, USA

*Correspondence to*: Isatis M. Cintron-Rodriguez (Isatis.cintron@gmail.com)

**Abstract.** Greenland's ice sheet mass loss rate has tripled since the mid-1950s in concert with sharply lowered albedo leading to increased absorption of solar radiation and enhanced surface melt. Snow and ice melt driven by solar absorption is enhanced by the presence of light absorbing particles (LAPs), such as black carbon (BC) and dust. Yet, the LAP impact on melt is poorly constrained, partly due to scarce availability of in-situ measurements. Here, we present a survey of snow properties and LAPs deposited in winter snow layers at five sites in southwest Greenland collected in May 2017. At these sites, BC and dust
concentrations were $0.62 \pm 0.35$ ng g-1 and $2.09 \pm 1.60$ μg g-1, respectively. By applying the SNICAR model, we show the LAP influence on albedo through the combined effect of surface darkening and snow metamorphism. While the LAP concentrations are low, they result in a 1.7% and 3.0% reduction in albedo within the visible spectrum for spring and summer, respectively. Past studies have shown that even minor LAP induced albedo reductions, if widespread, can have a large impact on the overall surface mass balance. SNICAR simulations constrained by our measurements show that LAP-snow aging
feedback reduce albedo reduction 4 to 10 times more than previously thought, therefore LAPs are likely a significant contributor to Greenland's accelerated mass loss. As far as we know, this is the first field study to consider the LAP impact on snow aging on the Greenland ice sheet.

## 1 Introduction

The Arctic is the fastest warming region in the world. Since the mid 1990s, Arctic near-surface air temperature has increased
twice as fast as the global average (IPCC, 2019; Richter-Menge et al., 2017; Notz and Stroeve 2016; Cohen et al., 2014; Chylek et al., 2009, Winton 2006). One of the many Arctic changes is accelerated mass loss from the Greenland Ice Sheet since the mid-1990s (Callaghan et al., 2011; Pritchard et al. 2009, Serreze et al., 2009; Comiso et al., 2008; Rignot et al. 2008; Stroeve et al., 2007, Groisman et al., 1994), which has become one of the main drivers of current sea level rise (Zhang et al., 2017). Over the past decades, the Greenland Ice Sheet melting contributed an average of $0.47 \pm 0.23$ mm yr[-1] sea level equivalent
(SLE) (1991-2015, van den Broeke et al., 2016) to the global sea level rise rate of 3 mm yr[-1](1993-2015, IPCC, 2019). Greenland's summer albedo, an important factor of surface melt, has decreased significantly since the 2000s (Riihelä



et al., 2019; He et al., 2013). Light-absorbing particles (LAP) deposited in snow contribute to this albedo reduction both by darkening the surface and hastening the melting season metamorphism. However, sparse measurements of LAPs from the Greenland ice sheet snow limits our understanding of the LAP and Greenland albedo reductions (Casey, 2017; Box et al., 35 2012; Hall, 2004).

Snow is an important climate regulator due to its high albedo in the visible to near infrared wavelengths coinciding with wavelengths with the most incoming solar energy (Warren, 1982). Snow albedo is influenced by a range of factors, including grain size, snow grain shape, water content, temperature and the presence of LAPs (Boy et al., 2019; Skiles et al., 2018; Ryan 40 et al., 2018; Adolph et al., 2017; Bullard et al., 2016; Skiles, 2015 Hadley and Kirchstetter, 2012; Hansen and Nazarenko, 2004; Warren and Wiscombe, 1985; Wiscombe and Warren, 1980). LAPs are often defined as the fraction of aerosols that can effectively absorb solar irradiance including black carbon (BC), dust, volcanic ash, and organic matter (Petzold et al., 2013; Moosmüller et al., 2012). LAPs in snow reduce albedo by direct absorption of incoming shortwave radiation causing surface darkening (Schneider et al., 2019; Skiles, 2018; Wiscombe and Warren, 1980).


Black carbon (BC), a major component in the soot produced by incomplete combustion of fossil fuels, biofuels, and biomass (Kirchstetter & Novakov, 2007), is the most efficient LAP at absorbing radiation in the visible range and it is the second most important anthropogenic warming agent after carbon dioxide ($CO_2$) with a total radiative forcing of +1.1 W m-2 (IPCC, 2013, Bond et al., 2013). Radiative forcing quantifies the influence of a given climatic factor on the Earth's energy balance (IPCC, 50 2007). The radiative forcing estimate of BC (+1.1 W m-2) accounts for all BC forcing mechanisms that include direct, cloud and cryosphere effects. Snow and ice BC direct radiative forcing (BC-snow forcing) has been estimated to be +0.65 W m-2, compared to $CO_2$ with +1.7 W m-2 and $CH_4$ with 0.95 +W m-2 (IPCC, 2019, Bond et al., 2013). Due to its short atmospheric lifetime (4 to 12 days) and radiative forcing, BC and other carbonaceous aerosols could reduce projected temperature increase by 0.5°C by 2050 while preventing millions of premature air pollution-related deaths and crop losses. (Samset et al., 2014; 55 Lee et al., 2013; UNEP & WMO, 2011).

Mineral dust aerosols are particles whose composition depends on the upper continental crust where they originated, with feldspar and quartz as dominating silicates (Lawrence et al., 2011). Deposition rate of dust at the poles is 6.5 million metric tons per year, which is above the global average (Lambert et al., 2008; Takemura et al., 2009). This rate is expected to increase 60 with global desertification and aridity as a result of climate change (Dai, 2011). Dust particles are removed and transported from the source by aeolian processes. During transport particles may fractionate and reduce the size of the dust particles transported over long distances to less than 20 to 30 µm (van der Does et al., 2018; Ryder et al., 2013; Kok et al., 2012). Evidence suggests that Arctic dust comes from Gobi and Taklaman deserts in Asia (38%), Saharan dust in Africa (32%), and local high latitude sources (27%) (Takemura et al., 2009). Besides darkening surfaces, dust promotes colonization of surface 65 algae yielding an increase in pigmented biomass and further albedo reduction (Yallop et al., 2012).



Previous studies demonstrate that LAP reductions on snow albedo in the Arctic are around 0.9% (Dang, 2017), and the combined influence of BC and enlarged snow grain size can decrease summer sea-ice albedo by 1.5 to 3.8% (Dou et al., 2016). Even such a small albedo change can significantly impact the ice sheet surface mass balance. Specifically, fresh snow albedo

reductions of only 1% can result in surface mass losses of 27 Gt a$^{-1}$ from the Greenland Ice Sheet (Dumont et al., 2014), which is equivalent to 16% of the average mass loss between 1991-2015 (van den Broeke et al., 2016).

Among the many unresolved questions to better understand Greenland's current and future contribution to global sea level are the impact of light absorbing particles (LAP) on snow albedo and surface melting. There is still uncertainty on the role of

different types of LAP (dust, black carbon, brown carbon, and organic matter) in snow albedo reductions. In a recent study, LAP concentrations in Greenland snow were found to be too low to significantly reduce albedo (Lewis et al. 2021). However, several feedbacks amplify LAP's direct albedo effect in snow including: 1) LAP-snow forcing where LAP energy absorption enhances grain growth (metamorphism) and therefore further darkening of snow (Schneider et al., 2019); and 2) Snow particle shape change has been observed to amplify this LAP-snow forcing (He et al., 2018) Furthermore, warmer temperatures

accelerate snow effective radius growth and the reconcentration of hydrophobic impurities near the snow surface following snowmelt (Flanner and Zender, 2006; Clarke and Noone, 1985; Wiscombe and Warren, 1980). This enhances the positive snow aging feedback loop, as part of the LAP-snow forcing, that accelerates metamorphism, increasing grain size and shape, allowing more solar irradiance absorption as a result of LAP surface reconcentration. While global climate models include snow radiative transfer computation, including parametrizations of snow physical characteristics (particle size, particle shape,

impurity load and solar zenith), the positive feedback of LAP on snow is often unaccounted in Greenland Ice Sheet measurements (Saito et al., 2019; He et al., 2018; Yasunari et al. 2011; Gardner and Sharp 2010; Marshall and Oglesby 1994). Our study area, southwest Greenland, had the fastest ice sheet mass loss from 2003 to 2013, as shown by Gravity Recovery and Climate Experiment (GRACE) measurements accounting for most of the sea level rise acceleration (Bevis et al., 2019; Chen et al., 2017). Hence, understanding the LAP impact on the ice sheet is important to understand its response to climate

change and potential mitigation strategies.

Here, we present observations of mass fractions of LAPs (BC and dust) in snow collected from five sites in southwest Greenland in 2017. We use the Snow, Ice, and Aerosol Radiative (SNICAR) model (Flanner et al., 2007) to estimate the magnitude of the contribution of LAPs to snow albedo reductions taking into account the impact of LAPs in snow

metamorphism (i.e snow grain size growth during aging).



## 2 Data and Methods

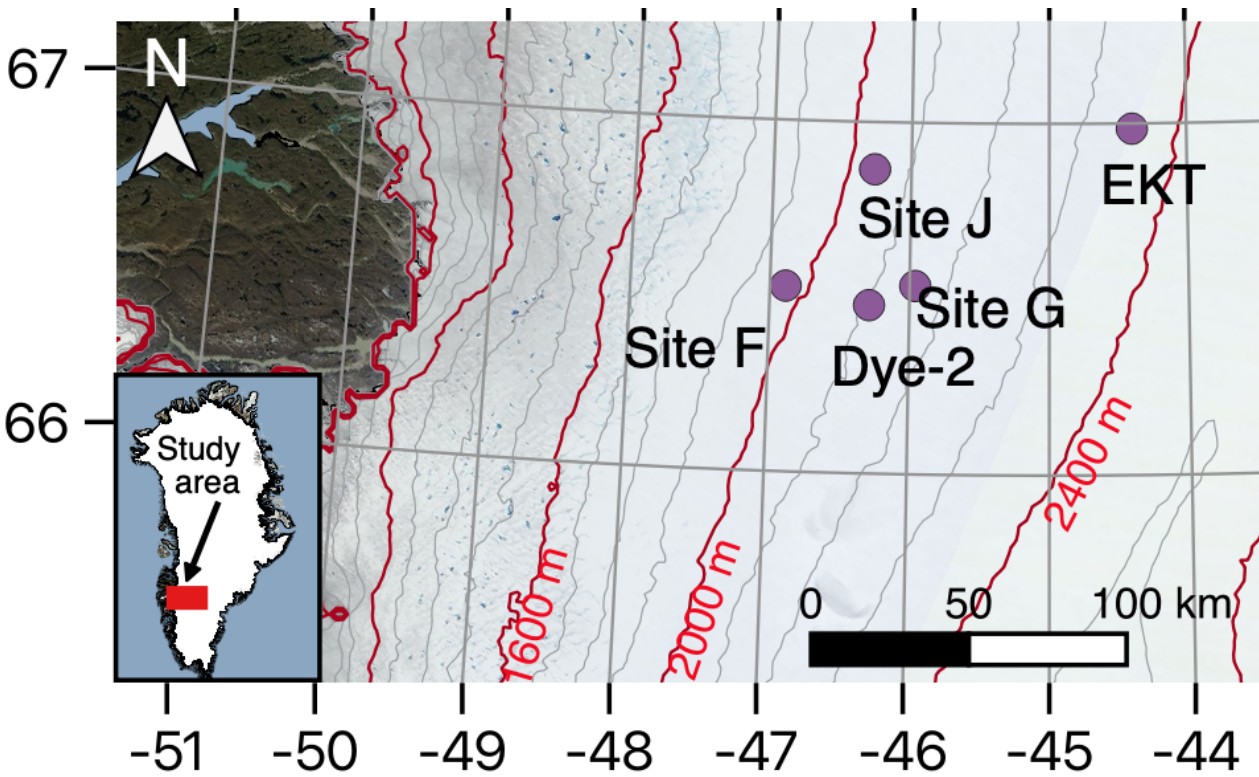

**Figure 1**. **Study area with locations of snow sampling sites (purple dots) in Southwest Greenland.** The ice sheet is white
with red superimposed elevation contour lines in m a.s.l, coordinates are latitude and longitudes (degrees east). The red box in
the inset map shows the extent of the larger map. Elevation contours in m a.s.l. are estimates based on the ArcticDEM 1 km
v.3.0 product by Polar Geospatial Center (Porter et al., 2018) adjusted with the EGM2008 geoid offset (Pavlis et al., 2012).
The background satellite map is from a composite of MODIS satellite imagery.

The physical and chemical composition of the annual snow layer was characterized using snow collected from five snow
sampling sites in southwest Greenland (Fig. 1, Table 1). The annual average snowfall and temperature at the five sites are
$334.2 \pm 11.6$ mm yr$^{-1}$ and $-17.03 \pm 1.17°$C (mean $\pm$ standard deviation 1949-2019) using Modèle Atmosphérique Régional
(MAR) reanalysis model version 3.11 (Fig 2). The data is from MAR version 3.11 run at a horizontal spatial resolution of 7.5
km using the ERA5 reanalysis dataset as forcing at the boundaries (Fettweis et al., 2017; Fettweis et al., 2020). The topography
is gentle and characterized by an almost flat surface at all five sites (Fig. 1).



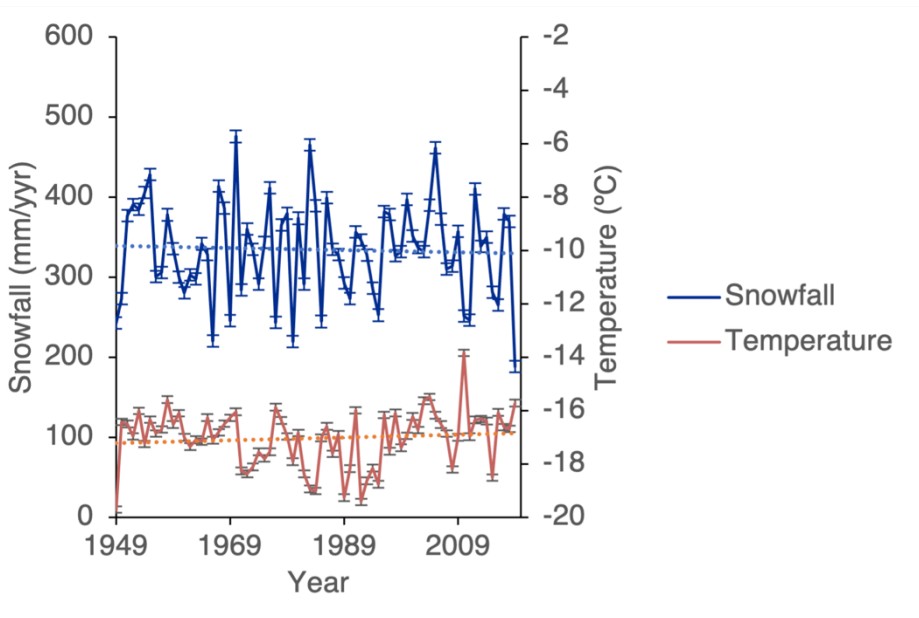

**Figure 2. Average annual water-year (October-September) snowfall and temperatures, from Modèle Atmosphérique Régional v.3.11 (MAR), and their standard deviations (error bars) for the five sampling sites of Southwest Greenland between 1949 and 2019.** The solid lines represent the annual averages, while the dotted lines indicate the linear trend.

### 2.1 Snow Sampling

A total of fifty-three snow samples were collected at the five locations between May 1-14, 2017 (Fig. 1). At each site, a snow pit was dug down to the depth hoar (usually around 1 meter below the surface). The depth hoar layer is easily distinguished by its large snow crystals that bond together poorly and marks the transition from recent snow (i.e. last winter snow), and firn (snow surviving at least one melting season). Two samples were collected every 0.15 - 0.2 m at six different depths from the surface to the depth hoar, except at Site G. To ensure samples remained uncontaminated, polypropylene gloves and non-particulating Tyvek suits were worn at all times during the sampling and handling. Pre-washed high-density polyethylene HDPE containers were used for collecting and storing the samples. Field blanks were prepared in bottles opened during field sample collection and brought empty to the laboratory. In the laboratory, field blank bottles were filled with ultrapure water, and handled identical as the snow samples bottles. Most samples partially melted before BC analysis but remained refrigerated. Environmental conditions such as precipitation in the last 48 hours, surface features, wind direction, and sky conditions were recorded at each site. Snow stratigraphy was recorded by measuring snow grain size, grain shape, density, and hardness at all sites. Snow temperature was measured with a digital thermometer (precision of ±0.1°C). Physical grain size and grain shape were determined with a snow card and hand magnifying lens solely stratigraphy purposes (optical grain size values described in Section 2.4). At three sites (Site J, Dye-2, and EKT) density was measured every 0.15-0.20 m following The National Aeronautics and Space Administration's guidelines (NASA, 2014). The snow density measurements were carried out using a



100 cm$^3$ or a 250 cm$^3$ SnowMetrics wedge (precision of ± 6 cm$^3$, Proksch et al. 2016), and digital scale with a precision of ±0.5 g. At the two remaining sites (Site F and Site G) density was measured in recent snow samples collected from a nearby drill hole. To collect drill hole samples, we used a mechanical ice-coring drill from the Ice Drilling and Design Office (IDDO). The drill set-up included a 2-m barrel (0.079 m diameter) connected to a Sidewinder (power hand drill and winch system,

Kyne and McConnell, 2007). However, this method to retrieve snow samples is less accurate than the using the SnowMetrics wedge, and often results in fragmented and incomplete density (Rennermalm et. al., in review). Indeed, we found Site F densities unrealistically low (< 250 kg m$^{-3}$ in 4 out of 5 snow layers), and they were discarded from this analysis and average densities from the other sites were used instead for site F.

**2.1 BC Measurements**

BC concentrations in the snow samples were determined using an extended range Single Particle Soot Photometer SP2 (Droplet Measurement technologies, Boulder, CO) coupled with a CETAC Marin 5 nebulizer (Wendl et al., 2014). Most previous studies of BC in arctic snow have used the Integrating Sphere/Integrating Sandwich spectrometer method (ISSW) or the thermal-optical transmittance (TOT) method (Polasensi et al., 2015; Forsstöm et al., 2013; AMAP et al., 2011; Doherty et al., 2010). However, these techniques have long standing identified uncertainties because of interference from coexisting non-BC

particles like mineral dust and filter undercatch (Doherty et al., 2016; Lim et al., 2014; Schwarz et al., 2013) and thus the SP2 methods were used here.

The SP2 is a laser-induced incandescence technique that combined with a nebulizer allows for the estimation of refractory BC (rBC hereby BC) in liquid samples (Kaspari et al., 2014; Schwarz et al., 2006; Wendl et al. 2014). The detection principle of

the SP2 is based on the heating to vaporization temperature (~4000 K) by an infrared intra-cavity laser and detection of laser-induced incandescence proportional to the mass of rBC (Stephens et al., 2003; Schwarz et al., 2006). This instrument is able to measure individual particles quantitatively and independent of their morphology and type of coating (Slowik et al., 2007; Schwarz et al., 2006; Stephens et al., 2003). One of the advantages of the SP2 method is the low sensitivity for particles other than rBC, providing a robust measurement of BC concentrations (Schwarz et al., 2006; Slowik et al., 2007; Moteki and

Kondo, 2007). The SP2 instrument coupled with a nebulizer allows the measurement of rBC concentration in liquid samples from melted snow in the case of this study (Bisiaux et al., 2012; Kaspari et al., 2011; McConnel et al., 2007). The CETAC Marin-5 nebulizer used in this study is characterized for its high efficiency and low sample consumption and is run under optimized operating parameters (Schwarz et al., 2013; Wendl et al., 2014, Mori 2016).

Snow samples for BC analysis largely melted during shipment to Central Washington University (CWU) and were near 0°C when they arrived. Because previous research demonstrated that refreezing samples could alter measured BC concentrations (Wendl el al., 2014), the samples were maintained refrigerated (4°C) until analysis. BC samples are typically kept frozen until



just prior to analysis, as BC losses due to particles adhering outside of the detection range of the SP2 and/or adherence to vial walls can result in a reduction in measured concentrations. However, these BC losses are most pronounced in samples that
have reached room temperature, and are minimal if the samples are kept cool (Wendl el al., 2014). Thus, our results may be slightly biased towards undersampling of BC.

The samples were sonicated for 15 minutes, then mixed with a magnetic stir bar, and pumped at 0.145 mL/min with a peristaltic pump to a CETAC Marin 5 nebulizer. The resultant aerosols were coupled to an extended range SP2. The particle size range
detected by the SP2 at CWU is 80-2000 nm mass-equivalent diameter for the incandescent signal, assuming a void-free BC density of 1.8 g cm$^{-3}$ (Moteki & Kondo, 2010). Reported BC concentrations are blank corrected based on deionized water (18.2 MΩ cm$^{-1}$), and BC losses that occur in the nebulizer are accounted by applying an external calibration using Aquadag standards (Wendl et al., 2014, Marquetto et al., 2020).

### 2.3 Dust Measurements

After the BC analysis was completed, the dry or insoluble microparticles mass (hereafter referred to as dry mass) was measured using a gravimetric filter procedure (EPA, 1998). Even though quartz filters are not ideal for dust measurements because of pore size uncertainty, they were selected as the filter medium to allow determination of other carbonaceous material in future work. Melted snow samples were filtered under a class 100HEPA (High Efficiency Particulate Air) clean bench using quartz filters previously baked at 650°C for 8 hours. After filtration, filters were allowed to dry in a Class 100 laminar flow bench.
Filters were weighted with an electronic balance (± 0.01 mg) before filtration, and 72 hours of equilibration period to allow adjustment to the constant conditions in a temperature and relative humidity (RH)-controlled room. Mass measurements were taken when room mean temperature was between 20-23 °C, with a maximum temperature variability under 2 °C over 24 hours, and a mean relative humidity between 30-40 % with less than 5 % variability over a 24-hour period.

This non-selective procedure, described above, results in a quantification of total dry mass, including all impurities deposited in snow, including carbonaceous particles, organic matter, and dust. To establish if our dry mass measurements are representative of the dust content, we use Iron (Fe) as a proxy for dust (Kaspari, 2014). Fe concentrations were determined through energy-dispersive X-ray fluorescence (ED-XRF) analysis, which is a nondestructive way to analyze filter samples. Fe was used as a proxy to constrain dust, following Kaspari et al. (2014), since alternatives such as ferric oxides tend to influence
mineral dust radiative forcing.





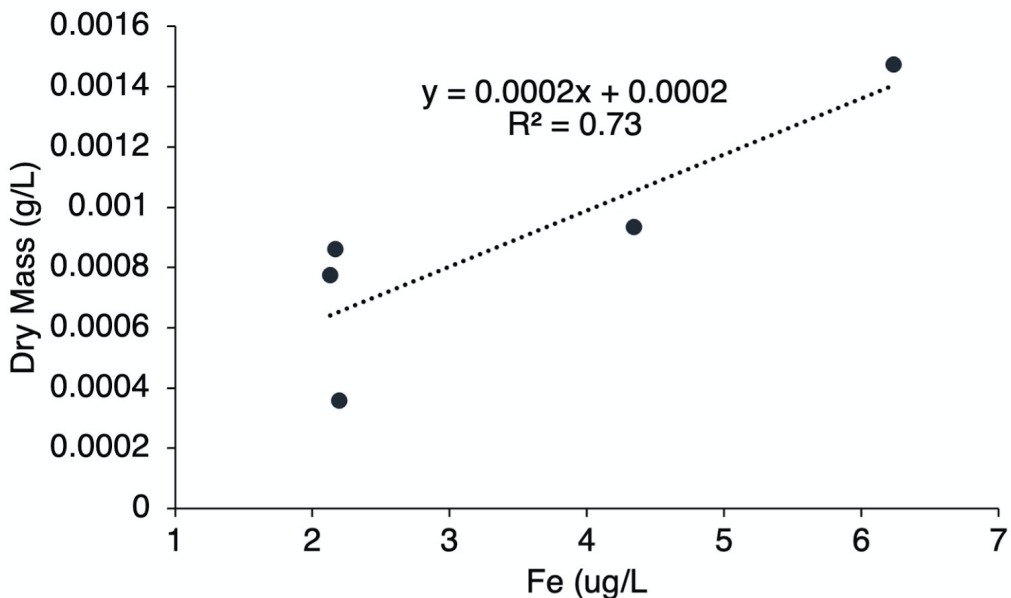

**Figure 3. Average Fe (ug L$^{-1}$) vs average dry mass (g L$^{-1}$) for each of the five sites (n=5).** The correlation coefficient between Fe and Dry Mass was 0.85 (p<0.001)

We found that average dry mass is statistically significantly correlated with average Fe concentrations for the five sites (n=5, p<0.001), which suggests that dust dominates the mass obtained by gravimetry (Kaspari, 2014). Given this significant correlation between Fe and dry mass, and the predominance of LAP dust in previous studies of Greenland snow (Ryan et al., 2018; Dumont et al., 2014; Drab et al., 2002), we conclude that our gravimetric measurements reflect dust concentrations.

**2.4 Estimates of albedo reductions with the SNICAR model**

We used the single-layer Snow, Ice, and Aerosol Radiation (SNICAR) model to estimate snow albedo resulting from combinations of deposited impurities, snow grain size, snow grain shape, and incident radiation (Flanner et al., 2007). SNICAR uses a two-stream multiple scattering approximation based on Toon et al. (1989) and Wiscombe and Warren (1980). We used SNICAR to simulate average albedo reductions in the visible spectrum (Flanner et al., 2007). The SNICAR model was run for clear-sky conditions for the sampling days and parameterized with snow density, stratigraphy and effective snow grain size

values from each site.

During the 2017 field campaign, snow grain size stratigraphy was collected following the NASA/US National Weather Service (NWS) protocol. Given that these manual measurements can't infer optical-equivalent grain size or effective snow grain radius ($r_{eff}$) needed for radiative transfer modeling we are using snow optical grain size data obtained from Arctic Data Center (Lewis,

2021). These values were retrieved from Field Spec 4 snow reflectance measurements at 1030nm carried out in southwest



Greenland between May 5 and May 31, 2017, its methodology is described elsewhere (Lewis et al., 2021). These data are assumed representative for our study sites since Lewi's sites are between 415 to 575 km away from the 2017 field sites used in this study and collected at a similar time. The average optical grain size $146.2 \pm 28.8$ was calculated from the values collected from the top 0.3 m, with a vertical resolution of 0.05m, of 10 snow pits located across a transverse in southwest Greenland (between -46.7306 °E, 70.5595 °N and - 49.4306 °E, 72.2001 °N).

These values of $r_{eff}$ are used in the SNICAR model to bracket the likely range of actual $r_{eff}$. Given that LAP accelerate snow metamorphism (Schneider & Flanner, 2017; Hadley & Kirchstetter, 2012), we ran the model using hexagonal grain shape for fresh snow modulations and spheroid shape for LAP-impacted snow to represent the positive feedback between black carbon and snow grain as the snow ages. This allows us to account for the potential increase in snow grain metamorphism owing to the extra solar energy absorbed by BC-laden snow. We use the Greenland dust properties determined by Polashenski et al., 2015 as inputs into SNICAR. The size distribution is assumed to be in the particle size bin of 2.5-5.0 μm based on previous glacial dust size range findings (Simonsen et al., 2019).

## 3 Results

### 3.1 Snow Properties

The average snow density was $369 \pm 42$ kg m$^{-3}$ for the four sites with reliable density (Site J, EKT, Dye-2, Site G), ranging from 308 to 472 kg m$^{-3}$ (Table 1, Fig. 4). In concert with high density, the temperature gradient affects dry snow metamorphism processes (Adams and Brown, 1983). The temperature-depth profile at the time of data collection was similar across sites, decreasing from -5 °C on average at the surface to -17.5°C at 0.80 m snow depth (Table 1). EKT had the greatest temperature gradient between the top and bottom layer, followed by Site J and Site B (Table 1).




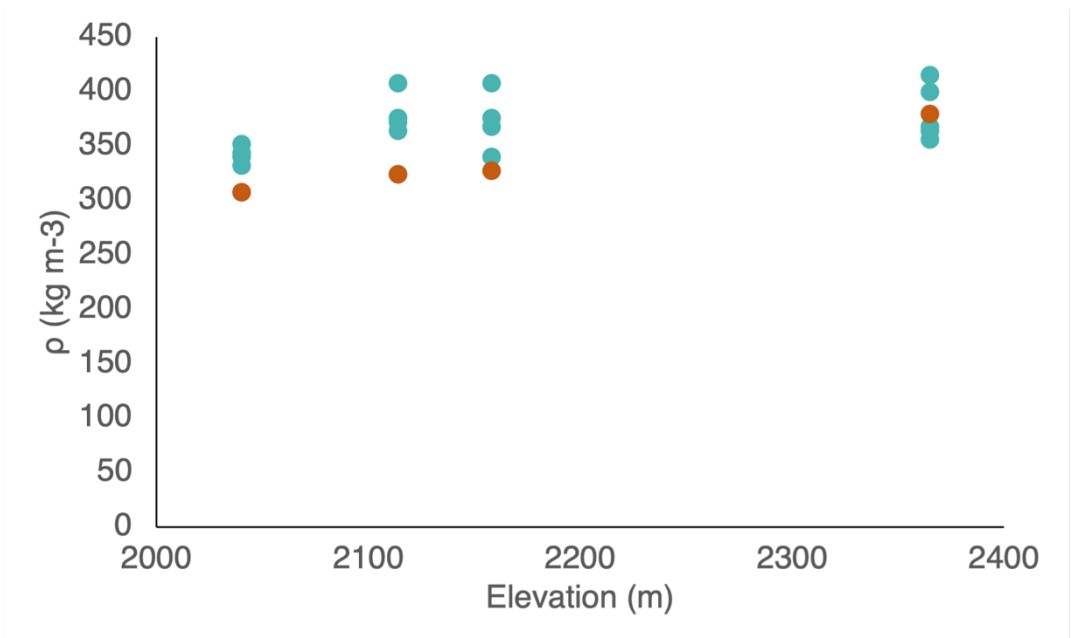

**Figure 4. Snow density (ρ) versus elevation.** Orange data points represent density (ρ) in the top layer (0-0.13 m) and the
blue data points represent the ρ in the rest of the snow layers (0.14-0.90 m).

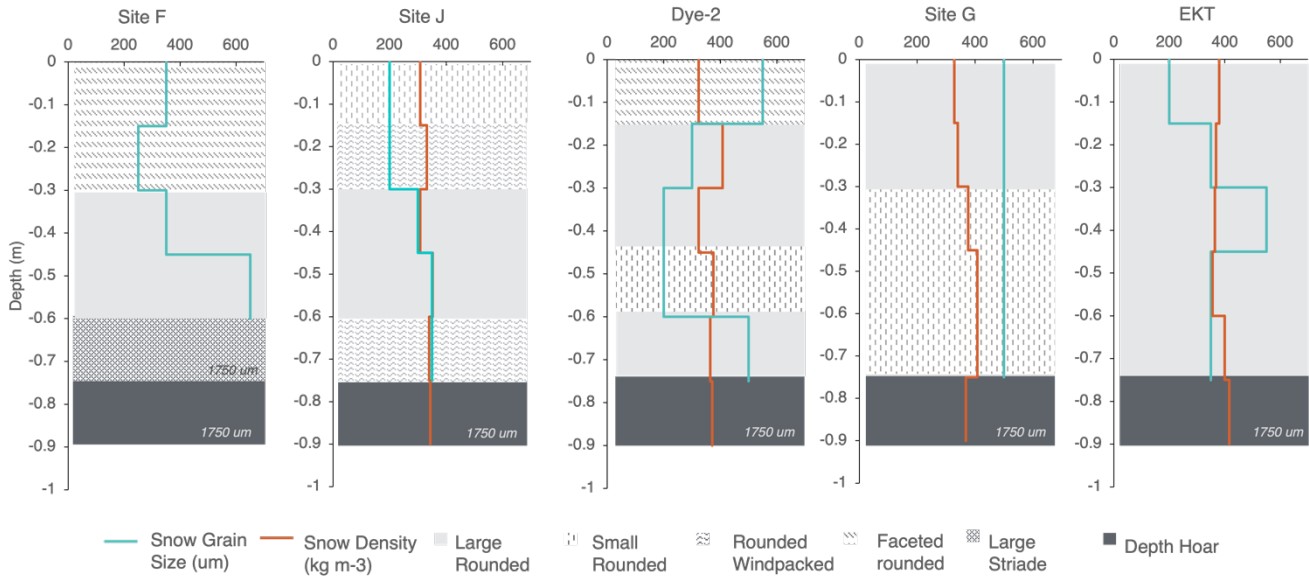

**Figure 5 Vertical profile of snow grain size, snow density (kg m⁻³) and grain shape stratigraphy observed at the five
sites in May 2017.** The sites are ordered by increasing elevation from left (Site F) to right (EKT). Variables were determined
in six depth intervals spanning 0.15-0.20 m at each site (hereafter layers).



The winter snow grain shape stratigraphy for the five sites are similar and consist of wind packed snow superimposing a depth hoar layer (Fig. 5). At all sites the top layer of snow is dominated by granular grain shape, with some sites having more varied grain shapes, including angular, faceted and granular, and rounded shapes (Table 1, Figure 5). Site J, Site F and EKT had three

different snow grain shapes while Site G and Dye-2  had two snow grain types. The grain size ranged between 0.20 and 0.67 mm above the depth hoar. The depth hoar, representing the lowest depth of the current year snowpack, occurred at a similar depth at all sites (0.72-0.82 m) with considerably larger grain sizes (mean was 1.59 mm) compared to the above lying snow (mean was 0.43 mm).

**Table 1** Summary statistics of snow properties and site descriptions Elevations were estimated ArcticDEM 1 km v.3.0 product by Polar Geospatial Center (Porter and others, Reference Porter2018) adjusted with the EGM2008 geoid offset as shown in Rennermalm et al., 2021.

| Variable | Site F | Site J | Dye-2 | Site G | EKT | Average |
|---|---|---|---|---|---|---|
| Elevation (m a.s.l.) | 1970 | 2040 | 2130 | 2160 | 2360 | - |
| Latitude (degrees N) | 66.5274 | 66.8650 | 66.4780 | 66.5351 | 66.9855 | - |
| Longitude (degrees E) | -46.8866 | -46.2651 | -46.2871 | -45.9591 | -44.3951 | - |
| Date sampled (2017) | May 14 | May 1 | May 11 | May 15 | May 7 | - |
| Snow temperature (°C) top layer | -6.0 | -11.0 | -9.5 | | -5.0 | -7.9 |
| Snow temperature (°C) bottom layer | -12.0 | -17.5 | -12.0 | | -17.5 | -13.2 |
| Snow grain size (mm) | 0.67 | 0.28 | 0.36 | 0.50 | 0.36 | 0.43 |
| Optical Grain Size (mm) | | | | | | |
| Snow density (kg m$^{-3}$) | - | 330 | 361 | 364 | 384 | 358 |
| Snow shape (top layer) | Faceted rounded grains | Small rounded grains | Faceted rounded grains | Large rounded grains | Large rounded grains | - |
| Depth hoar grain size (mm) | 1.75 | 1.75 | 1.20 | 1.50 | 1.75 | 1.59 |

**3.2 Light-absorbing particles concentrations**

The BC concentrations in individual snow samples ranged from 0.22 to 1.69 ng g-1. The BC concentrations in individual snow samples ranged from 0.22 to 1.69 ng g-1. BC concentrations decrease from the surface and then increase at subsequent layers most sites, except at Site J in which this pattern follows a peak between the 0.2-045 depth followed (Figure 6a, b c, e). Sites G and EKT present a slight divergence from this pattern. At EKT, BC decrease from the surface and remain low  except for a





local maximum at 0.4 m depth. Site G also follows this pattern although it has a small increase in BC at 0.45 m depth followed
by BC decrease in the subsequent layer (>0.60 m, Figure 6d).

BC concentrations in surface snow (0-0.15 m depth) ranged from 0.28 to 0.80 ng g$^{-1}$, and were highest at Site F followed by
EKT, Site G, Dye-2 and lastly Site J, likely as a result of a differing blowing snow disturbances among the sites. To calculate
total BC deposition over time at each site accounting for snow accumulation differences we use Delaney (2015):


$$BC_{tot} = BC_{avg} \; x \; \rho \; x \; D$$

where $BC_{tot}$ is total concentration in a given area of the snowpack, $BC_{avg}$ is the average BC concentration in the snowpit layers
(ng g$^{-1}$), $\rho$ is the average snow density, and D is the winter snow depth. $BC_{tot}$ range from 1.3 to 2.3 x 10$^4$ ng cm$^{-2}$ the five sites.
Site J had the largest cumulative BC concentrations with 3 x 10$^4$ ng cm$^{-2}$. The two sites with the lowest $BC_{tot}$ are Site G and
EKT, which also are at the highest elevations.







**Figure 6. BC (left, panels a - e) and dust mass (right, panels f - j) concentration with depth at the five sites.** Samples
were collected every 0.15 - 0.2 m and are hereafter referred to as the approximate mid-value of each depth interval (referred
to as layer in the text)

Dust concentration ranges from 0.04 to 2.81 µg g$^{-1}$, with an average 0.93 µg g$^{-1}$ (Figure 6). The dust concentration (Figure 6)
peaks in the layer centered at ~0.30 m for all sites except at Site F, where there is a peak at ~0.45 m layer. A second peak can
be found below 0.75 m, in the depth hoar layer, where dust concentrations are 1.52 ± 0.61. The highest dust concentration
among all individual samples is from Site J at the ~0.30 m layer with a value of 2.81 µg g$^{-1}$. This maximum value is followed
by Site F, Site G, EKT, and Dye-2. On individual samples, Dye-2 had the greatest concentration with 4.3 µg g$^{-1}$ but this was
found to differ significantly from other observations and was excluded from data set calculations.

**3.3 SNICAR simulations of snow albedo reductions**

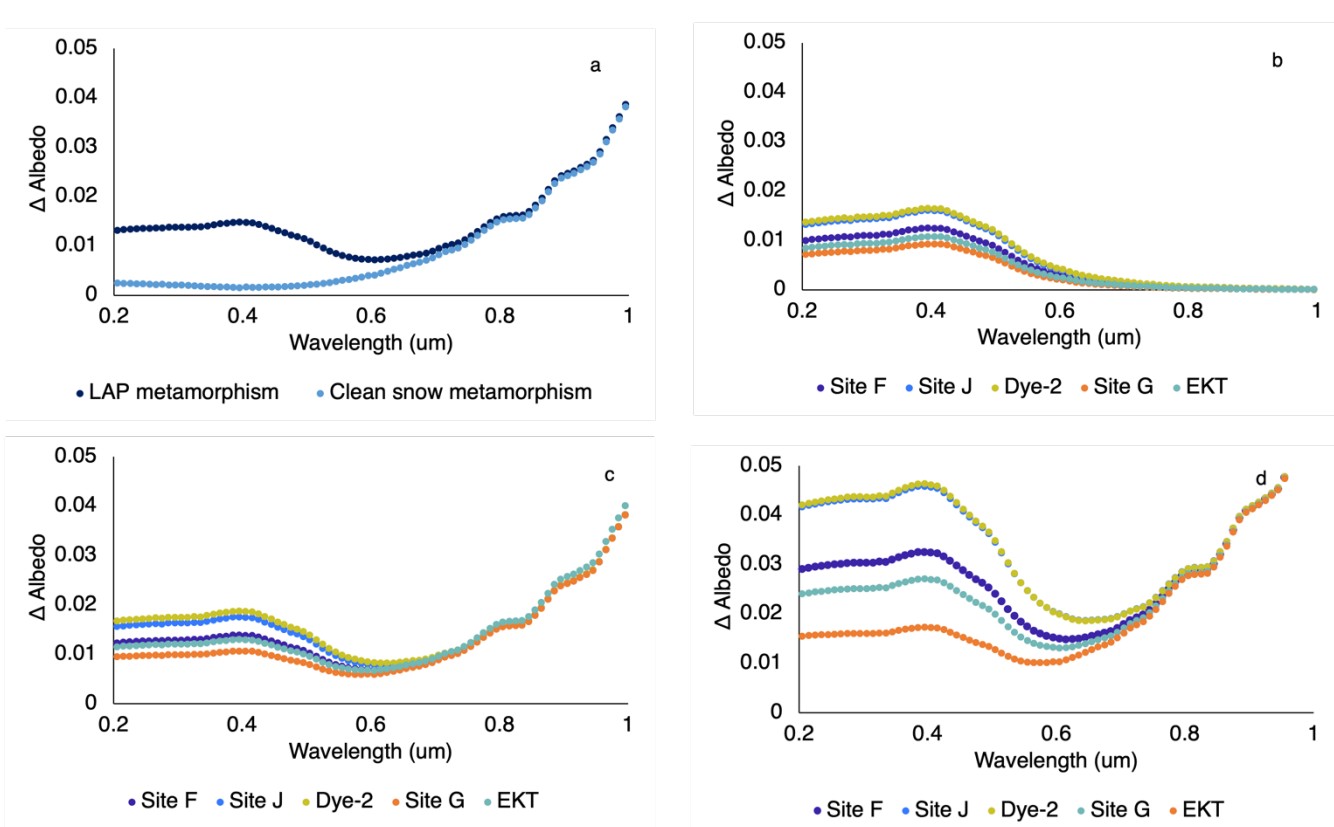

**Figure 7. SNICAR simulations of 2016-2017 southwest Greenland Ice Sheet albedo reductions with and without
metamorphism.** a) Spectral albedo changes in the visible spectrum due to snow metamorphism of clean snow (without
LAP influence) compared to LAP laden snow (average snow albedo reduction at all sites). b) Spectral albedo changes due to





the presence of LAP (with site-specific snowpack properties and light-absorbing particles (i.e. BC + dust). c) Spectral albedo changes due to LAP presence relative to clean snow (with SNICAR snow metamorphism modulation) d) Spectral albedo changes for the 2016 summer season using Warren observed snow properties for southwest Greenland Ice Sheet (personal correspondence). All spectral albedo changes represent the difference between the LAP-impacted snow and the clean snow simulated with the model using the same snow properties.


We ran the SNICAR model using the parameters listed in Table 2 for 2017 Spring albedo reductions. When comparing clean snow metamorphism vs LAP laden snow metamorphism we clearly see an influence in the visible region, that can't be attributed to the clean snow physical metamorphism alone (Figure 7a). Furthermore, snow albedo reductions are enhanced in the model simulations that take into consideration albedo reductions due to snow metamorphism (Figure 7c, Table 3). We

observe that Dye-2 and Site J have the greatest albedo reduction, while EKT presents the lowest albedo reduction. We observe that snow aging related BC, dust, and LAP combined albedo reductions are 2.6±0.5, 1.18±0.06, and 1.18±0.04, respectively, times greater than those related to non-SA simulations.

**Table 2.** List of site specific parameters used in SNICAR simulations. All runs were parameterized with clear-sky spectral
irradiance conditions for Summit Greenland and a snowpack thickness of 100 m. The model was run with hexagonal grain shape for Non-SA (SA = snow ageing) and spheroid grain shape for SA. We used surface values for BC and average snow pit values for dust as inputs to the model.

| Parameter | Site F | Site J | Dye-2 | Site G | EKT |
|---|---|---|---|---|---|
| Snowpack density (kg m$^{-3}$) | 345 | 330 | 361 | 368 | 386 |
| $r_{eff}$ (µm) | 146.7 | 146.7 | 146.7 | 146.7 | 146.7 |
| BC (ng g$^{-1}$) | 0.8 | 0.28 | 0.61 | 0.67 | 0.72 |
| Dust (2.5 – 5 um) (µg g$^{-1}$) | 0.87 | 1.34 | 1.37 | 0.56 | 0.70 |

**Table 3.** Albedo reductions for the wavelength band λ = 440 nm. . The model was run with hexagonal grain shape for Non-SA (SA = snow ageing) and spheroid grain shape for SA.

| | SA | | | Non-SA | | |
|---|---|---|---|---|---|---|
| Site | BC | Dust | Comb | BC | Dust | Comb |
| Site F | 0.003± 0.000 | 0.012± 0.001 | 0.013± 0.001 | 0.001± 0.000 | 0.011± 0.001 | 0.011± 0.001 |
| Site J | 0.002± 0.000 | 0.016± 0.002 | 0.016± 0.001 | 0.001± 0.000 | 0.014± 0.001 | 0.014± 0.001 |
| Dye-2 | 0.002± 0.000 | 0.017± 0.001 | 0.017± 0.001 | 0.001± 0.000 | 0.015± 0.001 | 0.015± 0.001 |





| | | | | | | |
|---|---|---|---|---|---|---|
| Site G | 0.003± 0.000 | 0.009± 0.001 | 0.010± 0.001 | 0.001± 0.000 | 0.007± 0.001 | 0.008± 0.001 |
| EKT | 0.003± 0.000 | 0.011± 0.000 | 0.012± 0.000 | 0.001± 0.000 | 0.009± 0.000 | 0.010± 0.001 |

**Table 4.** List of site specific parameters for the summer scenario SNICAR simulations. For these simulations, SNICAR was run with the BC and dust concentrations measured in the depth hoar layer, representing previous summer concentrations. We used previous measurements of snow effective grain size measured in the area in late-July 2010 provided by S. Warren (personal correspondence).

| Parameter | Site F | Site J | Dye-2 | Site G | EKT |
|---|---|---|---|---|---|
| Snowpack density (kg m$^{-3}$) | 500 | 500 | 500 | 500 | 500 |
| $r_{eff}$ (μm) | 550 | 550 | 550 | 550 | 550 |
| Black Carbon (ng g$^{-1}$) | 0.90 | 1.69 | 1.15 | 0.55 | 0.22 |
| Dust (2.5 – 5 μm) (mg g$^{-1}$) | 1.20 | 2.136 | 2.23 | 0.90 | 0.443 |

Using previous studies' summer measured snow grain size of 550 μm (Warren, personal correspondence) and the LAP concentrations in the depth hoar layer, detailed in Table 4, we parameterize the SNICAR model to estimate the spectral albedo reductions for summer 2016 (Figure 7d). As explained in the methods, the coarse-grained depth-hoar is a stratigraphic annual marker (Benson, 1959) and representative of past year. For the summer scenario, the sites display a 2.4-3.4% albedo reduction following a similar trend in which Site J and Dye-2 have the greatest change in albedo.







**Figure 8. Spectral albedo reductions due to LAP modelled using SNICAR at a solar zenith angle of 60° for different r$_{eff}$**
**and snow density combinations**. **Spectral albedo reductions due to LAP modelled using SNICAR at a solar zenith angle**
**of 60° for different r$_{eff}$ and snow density combinations**. Upper boundary (dotted lines) correspond to modelled albedo
reduction assuming BC (dust) coated and lower boundary values correspond to BC (dust) uncoated for a) r$_{eff}$ 50 μm, 60 kg m$^{-3}$
(fresh snow); b) 110 μm, 150 kg m$^{-3}$ (slightly aged); c) 200 μm, 250 kg m$^{-3}$ (settled snow); d) 350 μm, 375 kg m$^{-3}$ (wind packed
snow); e) 550 μm, 600 kg m$^{-3}$ (melting snow). This simulations are computed as the albedo of pure snow (hexagonal shape)
minus the difference of the albedo of pure snow (spheroid shape) the albedo of LAP- containing snow (spheroid shape). f)
Broadband albedo reductions for snow with LAP at r$_{eff}$ of 50, 110, 200, 350, and 550 μm.

Finally, we used the SNICAR to examine how the changing physical snow properties associated with aging of snow from
fresh, to settled, to windpacked affected the spectral albedo. We found that the radiative perturbation of LAPs increases with
snow aging. In melting snow, LAPs have double the impact on albedo reductions compared to fresh snow (Figure 8). We
computed the mean between the albedo reduction of coated and uncoated BC concentrations to illustrate the mean combined
effect of LAP-snow grain and shape feedback (Figure 8f). Snow grain growth and shape change amplify BC radiative
perturbation as the snow ages and melts. Thus, compared with fresh snow, BC concentration of 0.5 ug g$^{-1}$, and dust
concentration of 1.0 ug g$^{-1}$ causes an additional net albedo reduction of 0.01 - 0.15 depending on the impurity content in
melting snow under summer conditions. Furthermore, we find that the sensitivity to sulfate-coated BC (similar to BC
internally mixed in ice grains) increases with concentration of BC as shown in the difference between the upper (dotted) and
lower (solid) boundaries in Figure 8.

## 4 Discussion

### 4.1 Comparison with previous studies

Our findings of BC concentrations of 0.22 - 1.69 ng g$^{-1}$, are consistent with previous southeast Greenland snow studies, which
ranged from 0 to 5 ng g$^{-1}$ (Lewis et al., 2021, Mori et al., 2019; Stibal et al., 2017; Polashenski et al. 2015; Lim et al., 2014;
Carmagnola et al., 2013; Doherty et al., 2010; McConnell et al., 2007; Slater et al., 2002; Chylek et al., 1992; Clarke & Noone,
1985). While previous studies measured BC mass concentrations mainly using the Integrating Sphere/Integrating Sandwich
spectrometer (ISSW) and thermal-optical transmittance (TOT), the SP2 was used for this study. In contrast to the SP2, the
ISSW and TOT are prone to interferences of non-BC LAP leading to overestimation (Schwarz et al., 2013), and these methods
classify a larger portion of carbonaceous particles as BC. Thus, the higher BC concentrations of previous work relative to our
findings may be due to differences in methodology.





Given the hygroscopicity of BC, snow layers that have experienced melting tend to be enriched in BC explaining higher
concentrations in the surface and depth hoar layers. The lack of higher BC concentrations in the depth hoar layer at the highest
elevation site, EKT, may be due to the lesser surface melting at higher elevations leading to limited redistribution of
hydrophobic BC. We also observe decreasing BC with depth in the upper layers (0.10 - 0.45 m), except at Site J where BC in
the top layer was also the lowest among all sites (Figure 6). At Site J, meteorological observations indicate blowing snow
before the sampling time which suggests transport of fresh snow may have caused the lower surface concentrations of BC.
The deposition of fresh snow at Site J, also explains the downwards shift of the pattern with decreasing BC with depth, which
starts in the 0.2 m depth layer of Site J.

We found that dust concentrations averaged $0.93 \pm 0.59$ µg g$^{-1}$, with large vertical and spatial variability. Overall dust
concentration ranged from 0.04 to 4.39 µg g$^{-1}$, with the largest peak at 0.30 m depth in Dye-2. This peak may be explained by
the proximity of Dye-2 to an ice sheet aircraft runway maintained and used throughout May of 2017. Our dust values are in
the higher end of previous studies from different regions of the Greenland ice sheet (0.04 -1.90 µg g$^{-1}$, Simonsen et al., 2019,
Stibal et al., 2017; Polashenski et al., 2015; Kang et al., 2015; Aoki et al., 2014; Bory et al., 2003; Steffensen, 1997). High
variability is found in dust loads between seasons and years (Steffensen, 1997). Variations in dust estimation methodologies
(i. mass concentrations of crustal elements, non-sea salt $Ca^{2+}$ as a proxy and gravimetric procedures), season and location on
the Greenland ice sheet could have led to differences in dust loads. For example, gravimetric measures can only provide rough
estimates of dust concentrations. In this study, we use dry mass general properties and Fe as a proxy to estimate dust from dry
mass concentrations. Given that we did not analyze the mineral composition or water content in the dust matrix, our dust
concentrations may be overestimated due to the presence of organic matter.

### 4.2 Implications for regional climate modelling

Regional Climate Models (RCM) are one of the main tools to understand future Greenland Ice Sheet ice mass loss. RCMs,
such as the Regional Atmospheric Climate Model (RACMO), rely on parameterization of snow properties and light absorbing
impurities (e.g. Noel et al., 2018). RACMO has an energy balance snow metamorphism model used to calculate the surface
mass balance that includes the effect of BC on broadband snow albedo (Gardner and Sharp, 2010). The model estimates a 20%
increase in absorption of shortwave radiation for a snowpack with an average grain size of 1 mm, assuming a carbon loading
of 50 ng g$^{-1}$. Even though this loading is lower compared to older versions and achieve a better congruence between RACMO
and satellite observations of melt rates (Noel et al., 2018; van Angelen et al., 2012) this loading is higher than the measured
concentrations over the Greenland ice sheet that range between 0-5 ng g$^{-1}$. As highlighted in previous studies using RACMO,
the effect of BC varies with elevation, highlighting the importance of a broader understanding of the radiative effect of LAP
on the ice sheet to have better parameterization of regional climate models.





### 4.3 Implications on albedo

Based on our field measurements of snow properties and LAP abundance, we show that the spectrally weighted snow albedo can be reduced by $1.7 \pm 0.1\%$ in Southwest Greenland, taking into consideration the LAP snow aging positive feedback. This feedback is due to LAP enhanced melting, resulting in larger more spheroid shaped grains, which in turn facilitates solar radiation penetration and further grain growth and grain shape metamorphism. As far as we know, this is the first study to consider the LAP impact on snow aging on the Greenland ice sheet. When only considering albedo reductions due to LAP (not accounting for the snow aging feedback), the average broadband albedo reduction for the 2016-2017 BC concentrations is $0.4 \pm 0.1\%$ for the study area. This LAP-only impact is consistent with previous studies of BC impact on snow reflectance, which ranged from 0.2 to 0.5% albedo decreases (Dang et al., 2017; Dou et al., 2016). While our measured LAP concentrations were relatively low, we find larger albedo reduction than previous studies. This can be explained through the inclusion of LAP-snow aging positive feedback as an evolution of grain shape from hexagonal to spheroid. It has been shown that snow metamorphism can be enhanced by the presence of snowmelt that can produce rapid coarsening of ice grains (Brun, 1989). LAPs exert snowpack absorption influence in the visible wavelengths, while the isolated snow grain shape change (without LAP) influences mostly the albedo at the shorter wavelengths (Figure 7d). Therefore, after subtracting the clean snow grain shape change effect we can assume that all the observed absorption is due to the presence and snow metamorphism influence of LAP. Contrary to the findings of Lewis et al. 2021, we found that LAP have a significant role in the albedo reduction considering their role on accelerating snow metamorphism which amplifies LAP radiation perturbation (Schneider & Flanner, 2017; Hadley & Kirchstetter 2012).

Snow physical properties (temperature, density, and temperature gradient) influence the rate of aging, which occurs slower in cold snow (Oleson et al., 2010). Some prognostic parameters for snow metamorphism include $r_{eff}$, sphericity, and dendricity (Lehning et al., 2002). The SNICAR model uses a default snow aging scaling factor (SAF) of 1 (Oleson et al., 2010, Flanner and Zender, 2006). SAF is a multiplier on the instantaneous rate of $r_{eff}$ change dependent on dry and wet snow metamorphism and refreezing of melt water. Given the impact of snow metamorphism, we suggest using a factor of 1.18 to estimate the contribution of combined (dust and BC) LAP-induced albedo reduction due to the LAP-snow aging positive feedback at $\lambda = 440$ nm (Table 3). This factor would explain the difference in albedo reductions between SA and non SA scenarios at the maximum albedo reduction wavelength observed in Table 3. It must be noted that our snow grain shape and size aging approach does not consider effects of wind or refreeze cycles, which are also important factors in snow aging. Our results only apply for direct incident radiation given that half of our measurements were under clear-sky conditions. We note that overcast conditions in Greenland would render a slightly lower albedo perturbation than our results indicate.

In line with previous studies, interannual variability leads to greater albedo reduction over the summer or melting season under certain conditions, which coincides with much higher BC loads during summer than spring (Dou et al., 2016). The onset of

snowmelt accelerating albedo feedback and snow melting processes in summer are some factors that lead to high surface albedo interannual variability over the Arctic (Dou et al., 2012; Flanner et al., 2007). Our average albedo reductions are much higher for summer conditions than for spring ($3.0 \pm 0.4\%$), which suggest that the same impurities have a much greater effect, in summer as a result of the melt amplification due to LAPs presence and seasonal evolution of $r_{eff}$.G

## 4 Conclusion

This study presents direct measurements of light absorbing particles at five sites in southwest Greenland in spring 2017, where the snow represents accumulation for the water year. We measured the average BC concentration to 0.6 ng g$^{-1}$ which is at the lower end of previous studies from the region (ranging from 0 to 5 ng g$^{-1}$). In contrast, a commonly used regional climate model uses BC concentrations of 50 ng g$^{-1}$, possibly resulting in overestimation of surface melting. We measured the average dust concentration to $2.09 \pm 1.60$ µg g$^{-1}$. Using our observed BC and dust concentrations values and a model that considers snow metamorphism, we found that albedo is reduced by as much as $3.0 \pm 0.4\%$ in the summer and by $1.9 \pm 0.1\%$ as an annual average in contrast to fresh snow without LAP. These albedo reductions are 4 to 10 times larger than previous studies with similar LAP concentrations and suggests that LAPs have a greater impact on surface melting through snow metamorphism than previously thought. Constraining snow metamorphism processes is important to estimate LAP contribution to snowmelt, we demonstrate an order-of-magnitude improvement overestimates based on LAP direct effect only.

## 5 Author contributions

I C-R conceived the idea and designed field sampling. I C-R designed the project and wrote the majority of the manuscript with substantial contributions from Å. K. Rennermalm and S. Kaspari. S Z. Leidman collected the field data. All authors discussed the results and contributed to the final manuscript.

## 6 Competing interests

The authors declare that they have no conflict of interest.

## 5 Acknowledgement

I C-R was supported by the NSF Graduate Research Fellowship. Å K. Rennermalm and S Z. Leidman was supported by US National Science Foundations (NSF) (Grant OPP-1604058). S.Z. Leidman was also supported by the NSF Graduate Research Fellowship Program. ICR would like to thank Peter Kregsamer for access to the ED-XRF to perform Fe measurements. We



are grateful for the support by Polar Field Services for Greenland fieldwork logistics, and for the support by Regine Hock, Federico Covi, Clement Miege, Jonathan Kingslake, and Steven Munsell during the fieldwork. Maps and geographical
information science analyses were made with the open source QGIS software. The author would like to thank G. Lewis for his availability, responsiveness and support in providing the data of the OGS for running the SNICAR model. The MARv3.11 outputs were provided by Xavier Fettweis, University of Liege. Most importantly, we would like to express true gratitude to the land in which we performed our study, Kalaallit Nunaat (Greenland) and the Inuit peoples who have stewarded it through generations.

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
