# Peer review of "Light absorbing particles and snow aging feedback enhances albedo reduction on the Southwest Greenland ice sheet"

_The Cryosphere, 2022_

## Referee Comment (RC1)

**Comments on *Light absorbing particles and snow aging feedback enhances albedo reduction on the Southwest Greenland ice sheet* by Isatis Cintron-Rodriguez et al.**

**General comments**

The authors describe the snow and light absorbing particles (LAPs; black carbon and dust) measurements taken at five locations on the southwestern Greenland ice sheet in May 2017. They then assess the direct and indirect (through the effect of LAP-snow aging feedback) effects on snow albedo by using the SNICAR model.

The authors have presented a well-written and coherent manuscript on a topic that fits the scope of The Cryosphere well. Reporting and gaining understanding of LAP concentrations on the Greenland ice sheet is an important step towards improving snow albedo representation in (regional) climate models and towards more accurate projections of surface ice melting and sea level rise.

I have noted some minor comments below, the most important of which pertain to the discussion section. I would be great to see a bit more discussion of the limitations of the measurements, the methods, and the conclusions you can draw from this analysis. It would for instance be good to describe what effects other LAPs (not included in the measurements, such as algae, brown carbon, and volcanic dust) could have on the LAP-snow aging feedback and the albedo reduction in general.

**Specific comments**

General
Consider writing in the present tense instead of the past tense. It could make the text a bit more active.

L12: add algae to the list of LAPs.
L15: explain briefly what the SNICAR model is.
L20: do you mean *enhance* albedo reduction 4 to 10 times more?
L31: elaborate a little bit more on "Greenland's summer albedo". Do you mean of the ice sheet in general, of the snow, or of the ice?
L36: consider adding "global" before climate regulator.
L39: is temperature a direct regulator of snow albedo or does it change albedo by changing the grain size/shape and water content?
L43-44: remove "causing surface darkening". This aspect is covered in this sentence by "LAPs in snow reduce albedo".
L51-52: consider rewriting this sentence. It is not immediately clear to me what this means. Also make it clear that you are talking about $CO_2$ and $CH_4$ in the atmosphere, not in the ice/snow.
L62: elaborate on why the 20-30 micrometer is important.

L69-71: is it 16% of the average total mass loss (including dynamic losses)?

L76: which LAPs did Lewis et al. consider?

L79-83: make sure the leap from "snow effective radius growth" to "snow aging" is clear. Right now it is not clear what "snow aging" is.

L93-95: consider rewriting this sentence and elaborating a little bit on what SNICAR is and how you would use it to estimate this effect.

L108: MAR is a regional climate model, not a reanalysis model.

L120: what happened at site G?

L148-149: elaborate on what refractory BC is. Why can it be used as normal BC here?

L155: most of this is already mentioned before.

L157-158: what are the optimized operating parameters?

L163: what does "adhering to outside of the detection range" mean? What are the particles adhering to?

L180: is the 0.01mg the uncertainty?

L188-189: the first part of the sentence is already mentioned before.

L195: mention the correlation coefficient in the main text.

L203: what effects (due to LAPs) do you expect to see in the near infrared spectrum? Is it reasonable to not include this in your analysis? If so, show (with a reference) why.

L206-215: 415-575 km is a long distance and the meteorology and climatology can be very different between these locations. How do you know the situation is comparable?

L234: consider merging figures 4 and 5. Right now figure 5 has almost all the information embedded in figure 4, except for the elevation which you could add in the caption.

L242: is it winter snow grain shape or from May 2017?

L250: consider moving table 1 to earlier in the manuscript, when it is first referred to.

L266: does $BC_{tot}$ relate to the BC concentration in a vertical slice in the snow pack, given that the unit is ng cm$^{-2}$? If so, consider elaborating on that. It was not immediately clear to me.

L269: does average snow density refer to the density in each layer? If so, consider rewriting this sentence to make that clearer. It now seems like you take the average density of the entire layer.

L301: consider rewriting "that snow … albedo reductions are". It is not immediately clear to me what you mean here.

L350: elaborate on the BC concentrations in southeast and southwest Greenland. These regions are very different and might not be easily comparable. Given what you know about the sources and transport routes of BC to Greenland, elaborate on what differences/similarities would you expect to find between BC concentrations in southeast Greenland vs what you found in southwest Greenland? Take into account the general environment, elevation and topography of the sample locations in southeast and southwest Greenland. e.g. what side of the ridge of the GrIS each location is on and what is the general wind pattern from source to deposition location. Could local sources be affecting the BC concentrations (such as for dust at Dye-2)?

L405: briefly repeat the results of Lewis et al. 2021 here.

L413-415: move this to results. Also explain how you get the value 1.18.

L417-419: This statement is not immediately clear to me. If half of the measurements were under clear-sky conditions (and, thus, the other half under (partly) overcast conditions), would this not mean that your results do *not* solely apply to clear-sky conditions?

L421-422: what do you mean exactly with this? Are you talking about interannual variability of albedo reduction? If so, that cannot lead to greater albedo reduction over summer, it would the other way around.

Greater albedo reduction in summer leads to interannual variability of albedo reduction. Or do you mean interannual variability of another variable? If so, make clear what you are referring to here.

L423: elaborate what you mean with snow melting processes as a factor leading to albedo variability.

L425: does the $3.0 \pm 0.4\%$ refer to the difference between summer and spring albedo reduction? Consider mentioning both average albedo reductions for spring and summer to show the difference.

L425: can part of this albedo reduction difference between summer and spring not be attributed to higher LAP concentrations between the seasons?

L429: what is the water year?

**Technical corrections**

General
Check double spaces and units throughout the manuscript.

L15: put -1 in superscript in ng g$^{-1}$. Also check rest of the manuscript for this.

L29: turn "the Greenland Ice Sheet melting" into "melting from the Greenland Ice Sheet".

L32: turn "deposited in snow" into "deposited on snow".

L33: turn "melting season metamorphism" into "snow metamorphism induced by melting".

L34: turn "LAP" into "LAPs".

L47: turn "is the most efficient LAP at absorbing radiation" into "is the LAP most efficient at absorbing radiation".

L53: turn "reduce projected" into "reduce the projected".

L61: add comma after "transport".

L67: turn "LAP reductions on snow albedo" into "snow albedo reductions due to LAPs".

L70: consider using "yr$^{-1}$" instead of "a$^{-1}$". The latter is usually used for time backwards.

L77: turn "LAP's" into "LAPs".

L78: turn "Snow" into "snow".

L79: add period after reference.

L100: turn "longitudes" into "longitude".

L110: remove "gentle and".

L118: consider rewriting to "from recent snow (i.e. snow from last winter) to firn (snow surviving at least one melting season)."

L124: turn "identical as" into "identically to".

L127: remove plusminus sign. Same for L131 and L132

L128: add "for" before "stratigraphy".

L139: turn "2.1" into "2.2".

L142: capitalize "arctic".

L148: add comma's after "that" and "nebulizer".

L172: add "for" after "accounted".

L182: turn "room mean" into "mean room".

L185: consider attaching to previous paragraph and removing the white line. Also remove "described above".

L193: make sure the superscript in L$^{-1}$ is of approriate size.

L193: add closing parenthesis in x-label.

L211: remove ", its methodology is described elsewhere".

L212: turn "Lewi's" into "Lewis'".

L238: add units of snow grain size in caption.

L250: add period after descriptions.

L251-252: check format of references.

L256: add "at" after "layers".

L257: turn "except at … follow (Figure 6a, b, c, e)" into "except at site J which exhibits a peak in BC concentrations at 0.2-0.45 m depth (Figure 6)".

L263: turn "differing blowing snow disturbances" into "difference in snow accumulation".

L266: turn "x" in equation into dot operator symbol.

L269: add "concentrations" after "$BC_{tot}$".

L269: add "at" before "the five sites".

Figure 7: change "u" into "μ" in the x labels.

Table 2: change "u" into "μ" in the dust row.

L293-294: move this sentence to the beginning of the figure caption.

Figure 8: reverse the order of the legend entries so it matches the order of the lines.

L302: introduce the abbreviation "SA" in the main text.

Figure 8: change "u" into "μ" and put "-1" in superscript in the x labels.

L329-331: double sentence.

L334: turn "this" into "these".

L335: add "and" after "shape)".

L350: remove comma before "are".

L365: remove comma before "also".

L365: remove "s" in "downwards".

L368: add comma after "overall".

L374: check "i.". Should this be "i.e."?

L390: turn "on" into "for".

L410: turn "reff" into "$R_{eff}$".

L412: turn "reff" into "$R_{eff}$".

L426: remove "G".

L426: turn "reff" into "$R_{eff}$". Or rewrite everywhere to "$r_{eff}$". Both are okay, as long as you are consistent.

L436: use a period instead of a comma, or rewrite sentence.

L437: remove "overestimates"?

---

## Referee Comment (RC2)

**Review for TC Cintron-Rodriguez et al Light absorbing particles and snow aging feedback enhances albedo reduction on the Southwest Greenland ice**

**General comments**

This manuscript describes a set of observational data taken in the southwest ablation zone of the GrIS and applies SNICAR to estimate the total impact LAP and snow metamorphism have on albedo. These measurements include dust and BC concentration, snow density, and snow grain size and shape. This data is useful for constraining the concentration of LAPs and snow properties in southwestern Greenland. This set of observational data is interesting and useful. This paper is generally well written and is an interesting concept. However, I think this paper needs major revisions and the methodology needs to be revisited. Overall, it is unclear to me how properties used, assumptions made with regard to grain shape and size, and the calculation of total change in albedo and impact of LAP on snow metamorphism is unclear.

My most substantial suggestions are (in no particular order)

- 1. Use the multi-layer SNICAR-ADv3 or 4 (both are available online). While properties lower in the snow pack will only minimally influence the albedo, it would be beneficial to utilize the in situ measurements with depth. It would also be nice to see the modeled output albedo for the simulations constrained by measurements, with the caveat that the true albedo is not known.
- 2. Address the influence of grain size with regard to the results presented in figure 7, the manuscript states that both grain size and grain shape change with snow metamorphism and both have implications on albedo. However, only snow grain shape is addressed with the measurements taken. It is also important to provide justification of why hexagonal grains are used to represent fresh snow and spheroid grains are used to represent aged snow.
- 3. It is unclear what the relationship between LAP and snow metamorphism is, for example, what concentration of LAP justifies a transition from hexagonal shaped grains to spheroidal grains? How much of the grain growth and shape change can we attribute to LAPs rather than environmental variables such as temperature?
- 4. Please provide a more thorough justification for the snow properties and LAP concentrations used to simulate the results presented in figure 8. These results are all theoretical as there is no justification of the snow conditions used to simulate the albedo differences.
- 5. There needs to be significant clarifications within the methods for how the change in albedo is calculated. It is unclear how the total changes and % changes in albedo are calculated, and as a result these results are not easily reproducible.
- 6. I think the introduction could include more discussion about other measurements and how the measurements described here are different and useful (the measurements were taken with depth, you measure the grain size, shape, and density of the snow).

I have other specific comments below.

Specific comments

- 1. Line 19-20: "SNICAR simulations constrained by our measurements show that LAPsnow aging feedback reduce albedo reduction 4 to 10 times more than previously thought," Please add more nuance than "more than previously thought". For example, are you comparing to a study that also used measurements and RT modeling? Then say "more than previous studies have estimated" It's also important to elaborate on how this range was calculated in your methods / results.
- 2. Line 33-34: "However, sparse measurements of LAPs from the Greenland ice sheet snow limits our understanding of the LAP and Greenland albedo reductions" it could be useful to briefly discuss past measurement campaigns and different LAP data and to move your discussion of previous measurements to the introduction and then in the results you can say your measurements compared well and explain the possible discrepancies (Bøggild et al., 2010; Wientjes et al., 2012; Wientjes and Oerlemans, 2010; Cook et al., 2020; Onuma et al., 2019).
- 3. Line 53-55: "BC and other carbonaceous aerosols could reduce projected temperature increase by 0.5°C by 2050 while preventing millions of premature air pollution-related deaths and crop losses. (Samset et al., 2014; Lee et al., 2013; UNEP & WMO, 2011)." The removal of BC?
- 4. Lines 61-64: "During transport particles may fractionate and reduce the size of the dust particles transported over long distances to less than 20 to 30 μm (van der Does et al., 2018; Ryder et al., 2013; Kok et al., 2012). Evidence suggests that Arctic dust comes from Gobi and Taklaman deserts in Asia (38%), Saharan dust in Africa (32%), and local high latitude sources (27%) (Takemura et al., 2009)." This doesn't seem very relevant. It would be more useful to provide background on relevant measurements in this region of the ice sheet.
- 5. Snow and ice algae and the exposure of bare ice/crustal surfaces have been found to be very strong albedo reducers, I think your discussion should include some mention of these other mechanisms (Chevrollier et al., 2022; Tedstone et al., 2020; Ryan et al., 2019)
- 6. Line 83-86: "While global climate models include snow radiative transfer computation, including parametrizations of snow physical characteristics (particle size, particle shape, impurity load and solar zenith), the positive feedback of LAP on snow is often unaccounted in Greenland Ice Sheet measurements (Saito et al., 2019; He et al., 2018; Yasunari et al. 2011; Gardner and Sharp 2010; Marshall and Oglesby 1994)." He and others have done significant work to improve this representation, it is likely worth mentioning their work (He, 2022; Hao et al., 2022). It is also unclear if this sentence is about the importance of this representation in model or measurements. Please adjust.
- 7. Line 93: "Snow, Ice, and Aerosol Radiative (SNICAR) model (Flanner et al., 2007)" It is unclear which version of SNICAR is used in this study. The 2007 version does not allow for various grain shapes. There are multiple newer versions of this model that have been recently released that allow for multi-layer simulations and various grain shapes (Whicker et al., 2022; Flanner et al., 2021; Dang et al., 2019). These newer versions do not utilize the Toon 1989 solving method. Please clarify in the methods.
- 8. Line 200: "We used the single-layer Snow, Ice, and Aerosol Radiation (SNICAR) model to estimate snow albedo" It would be more interesting if you used a multilayer approach

using SNICAR-ADv3 or SNICAR-ADv4 and utilized your snow measurements with depth

- 9. Line 216: "These values of reff are used in the SNICAR model to bracket the likely range of actual reff." There was no use of a range of reff in the results? If you were going to use  $146.2 \pm 28.8$  as a range you would need to run 3 simulations, one for reff of 146.2-28.8, one for 146.2, and one for 146.2+28.8. This is also different from the values reported in table 2.
- 10. Figure 5 is great, it could be interesting to try and overlay [dust/BC] so we can see all the snow properties in one figure, or at least improve the dust/BC figure to look similar to figure 5.
- 11. Figure 7 caption: Please find a way to more clearly represent what delta albedo you are showing. It is unclear which figures include changing snow grain shape and which only include the influence of LAPs. For example use a more descriptive ledged like "spheroid grains w/ LAPs hexagonal plate w/ LAPs". The last line of the caption makes it seem like the same grain size and shape was used for all simulations "All spectral albedo changes represent the difference between the LAP impacted snow and the clean snow simulated with the model using the same snow properties." It is also unclear which snow properties are used in 7d, please refer to the corresponding table.
- 12. Table 2: The same grain size was used for all (both SA and non-SA) simulations? If that is the case, these simulations don't capture the full influence of snow metamorphism. Why is the average dust concentration used but only the surface BC used in the simulation? Also the effective radius 146.7 is not an option within SNICAR please adjust. Please also include all SNICAR parameters so these results can be recreated. For example, make a SA and non-SA column with all model input parameters (including the varying LAPs) so these simulations can easily be recreated and the differences between the simulations are clear. Please also apply this technique to the SNICAR parameters used for the figure 8 simulations.
- 13. Line 300-303: "We observe that snow aging related BC, dust, and LAP combined albedo reductions are 2.6±0.5, 1.18±0.06, and 1.18±0.04, respectively, times greater than those related to non-SA simulations." These albedo reductions are simulated, not observed. It is also not clear which simulations these reductions are based on. Why is the BC only reduction larger than that of the LAP combined albedo reduction?
- 14. Line 320: "Using previous studies' summer measured snow grain size of 550 μm (Warren, personal correspondence) and the LAP concentrations in the depth hoar layer, detailed in Table 4, we parameterize the SNICAR model to estimate the spectral albedo reductions for summer 2016 (Figure 7d) " Table 4, The density values in table 4 are not representative of the summer 2016 values as the snowpack has been compacted by melting, refreezing, and fresh snowfall. While the density doesn't influence the albedo of snow in SNICAR, it is likely worth mentioning that this density is greater than that of the 2017 snow density and making sure the reader understands that this will not have an impact on the delta albedo calculations. Please explain why using the snow grain radius from Warren is better than following the method used for the 2016 grain size, the 550 μm grain size is much larger than the 146.2 um grain size estimate.
- 15. Line 336: "f) Broadband albedo reductions for snow with LAP at reff of 50,110,200,350,and550μm." This simulation only includes BC, correct? Please adjust the legend accordingly.

- 16. Line 331-332: "Upper boundary (dotted lines) correspond to modelled albedo reduction assuming BC(dust)coated and lower boundary values correspond BC (dust) uncoated" this seems like you're getting the same albedo reduction for BC and dust? As you're using one line to represent different concentrations of both. Are you using both BC and dust in one simulation? I think your units for dust concentration are incorrect on the figure. Are you using milligram/gram (mg/g) or microgram/gram (ug/g)?
- 17. 332-334: "a)reff 50µm,60kgm-3 (fresh snow); b) 110 µm, 150 kg m-3 (slightly aged); c) 200 µm, 250 kg m-3 (settled snow); d) 350 µm, 375 kg m-3 (wind packed snow); e) 550 µm, 600 kg m-3 (melting snow)." There is no justification for why these specific snow properties are chosen for aged/settled/wind packed/melted snow. Please include citations and discussion for why these snow properties were chosen for each snow state. Please also justify the concentrations of dust and BC you are using. In your measurements, I see a max dust concentration of ~3ug/g and a max BC concentration of ~2ng/g, but in your figure 8 simulations you are using up to 200ug/g (I am assuming mg/g on the figure is a typo) of dust and 1500 ng/g of BC. The use of such high LAP concentration inflates the difference in albedo and these concentrations seem unjustified based on the measurements presented here and your comparison to previous measurements.
- 18. Line 334:335: "This simulations are computed as the albedo of pure snow (hexagonal shape) minus the difference of the albedo of pure snow (spheroid shape) the albedo of LAP- containing snow (spheroid shape)." It is unclear what differences are being represented here, it might be more clear if you also show the SNICAR output albedo with sufficient legends to be more clear about which albedos you are differencing and/or write out an equation. It sounds like you are differencing three albedo simulations, which does not seem correct.
- 19. Line 340: "LAPs have double the impact on albedo reductions compared to fresh snow (Figure 8)." Please include quantitative results and range, this seems like it is only true for very large LAP concentrations? If so, this result is contingent on the justification of both snow properties and LAP concentrations used in figure 8.
- 20. Line 345: "Thus, compared with fresh snow, BC concentration of 0.5 ug  $g^{-1}$ , and dust concentration of 1.0 ug  $g^{-1}$  causes an additional net albedo reduction of 0.01 0.15 depending on the impurity content in melting snow under summer conditions." Why does this depend on impurity content? Isn't this result for a simulation with BC concentration of 0.5 ug  $g^{-1}$ , and dust concentration of 1.0 ug  $g^{-1}$ ? Are you comparing fresh snow with LAP to aged snow with LAP?
- 21. Line 398-399: "While our measured LAP concentrations were relatively low, we find larger albedo reduction than previous studies." Do you have citations for these previous studies? I'm not sure this is fair to say because no in situ albedo measurements were taken. It is important to note alongside this discussion of the impact on albedo that the uncertainty associated with this albedo is unknown because no in situ albedo measurements were taken. Also, are theses albedo reductions based on theoretical snow properties and LAP concentrations (ie not measured values in figure 8)? If so that major caveat needs to also be mentioned.
- 22. Line 403: "while the isolated snow grain shape change (without LAP) influences mostly the albedo at the shorter wavelengths (Figure 7d)" What wavelengths and by how much? Changes to grain shape and size have the strongest impact in the NIR, so seems

counterintuitive to not include this region of the spectra in your analysis. Figure 7d does not show the impact of changing snow grain shape.

- 23. Line 404-405: "Therefore, after subtracting the clean snow grain shape change effect we can assume that all the observed absorption is due to the presence and snow metamorphism influence of LAP" This absorption is not observed, it was simulated. It is also unclear what you are trying to say here
- 24. Lines 405-407: "Contrary to the findings of Lewis et al. 2021, we found that LAP have a significant role in the albedo reduction considering their role on accelerating snow metamorphism which amplifies LAP radiation perturbation (Schneider & Flanner, 2017; Hadley & Kirchstetter 2012)." I'm not sure this is fair to say, the role LAP's play in accelerating snow metamorphism is not clear from this study. You do not calculate how the LAPs are influencing the grain shape, you simulate the impact the combined effect of LAPs and snow metamorphism have on albedo. Can you suggest the rate of metamorphism based on only LAPs (not the influence of temperature or general snow aging)?
- 25. Line 417-419: "Our results only apply for direct incident radiation given that half of our measurements were under clear-sky conditions. We note that overcast conditions in Greenland would render a slightly lower albedo perturbation than our results indicate." Why is this relevant? Wouldn't the sky conditions only be relevant for albedo measurements?
- 26. Lines 432:434: "Using our observed BC and dust concentrations values and a model that considers snow metamorphism, we found that albedo is reduced by as much as 3.0 ± 0.4% in the summer and by 1.9 ± 0.1% as an annual average in contrast to fresh snow without LAP" It seems as though in the simulations where measured dust and BC are used (figure 7) only changing snow grain shape is accounted for, there is no grain growth, so it doesn't seem accurate to say "the model considers snow metamorphism".
- 27. Lines 434-436: "These albedo reductions are 4 to 10 times larger than previous studies with 435 similar LAP concentrations and suggests that LAPs have a greater impact on surface melting through snow metamorphism than previously thought" Please explain how the 4 to 10x is calculated.

**Technical corrections**

- 1. Line 79: missing period "(He et al., 2018) Furthermore"
- 2. Line 54: extra period "crop losses. (Samset et al., 2014; Lee et al., 2013; UNEP & WMO, 2011)."
- "Figure 7. SNICAR simulations of 2016-2017 southwest Greenland Ice Sheet albedo reductions with and without metamorphism" are these for 2016 or 2017? Text says measurements from 2017 "We ran the SNICAR model using the parameters listed in Table 2 for 2017 Spring albedo reductions" but the figure title says 2016-2017
- 4. "Table 4. List of site specific parameters for the summer scenario SNICAR simulations. For these simulations, SNICAR was run with the BC and dust concentrations measured in the depth hoar layer, representing previous summer concentrations." Please say the year (ie 2016) rather than "previous summer"
- 5. Lines 329-331: "Figure 8. Spectral albedo reductions due to LAP modelled using SNICAR at a solar zenith angle of 60° for different reff and snow density combinations.

Spectral albedo reductions due to LAP modelled using SNICAR at a solar zenith angle of 60° for different reff and snow density combinations" title of figure included twice

- 6. Figure 8F needs a y axis label
- 7. Please add the effective radius and snow density used for each subplot within each subplot.
- 8. Be consistent with the units of grain size, some are in mm and some are um. Please generally be consistent with unit notation.

**Relevant Citations:**

Bøggild, C. E., Brandt, R. E., Brown, K. J., and Warren, S. G.: The ablation zone in northeast Greenland: ice types, albedos and impurities, Journal of Glaciology, 56, 101–113, https://doi.org/10.3189/002214310791190776, 2010.

Chevrollier, L.-A., Cook, J. M., Halbach, L., Jakobsen, H., Benning, L. G., Anesio, A. M., and Tranter, M.: Light absorption and albedo reduction by pigmented microalgae on snow and ice, J. Glaciol., 1–9, https://doi.org/10.1017/jog.2022.64, 2022.

Cook, Hodson, A. J., Gardner, A. S., Flanner, M., Tedstone, A. J., Williamson, C., Irvine-Fynn, T. D. L., Nilsson, J., Bryant, R., and Tranter, M.: Quantifying bioalbedo: a new physically based model and discussion of empirical methods for characterising biological influence on ice and snow albedo, The Cryosphere, 11, 2611–2632, https://doi.org/10.5194/tc-11-2611-2017, 2017.

Cook, J. M., Tedstone, A. J., Williamson, C., McCutcheon, J., Hodson, A. J., Dayal, A., Skiles, M., Hofer, S., Bryant, R., McAree, O., McGonigle, A., Ryan, J., Anesio, A. M., Irvine-Fynn, T. D. L., Hubbard, A., Hanna, E., Flanner, M., Mayanna, S., Benning, L. G., As, D. van, Yallop, M., McQuaid, J. B., Gribbin, T., and Tranter, M.: Glacier algae accelerate melt rates on the south-western Greenland Ice Sheet, The Cryosphere, 14, 309–330, https://doi.org/10.5194/tc-14-309-2020, 2020.

Dang, C., Zender, C. S., and Flanner, M. G.: Intercomparison and improvement of two-stream shortwave radiative transfer schemes in Earth system models for a unified treatment of cryospheric surfaces, The Cryosphere, 13, 2325–2343, https://doi.org/10.5194/tc-13-2325-2019, 2019.

Flanner, M. G., Arnheim, J. B., Cook, J. M., Dang, C., He, C., Huang, X., Singh, D., Skiles, S. M., Whicker, C. A., and Zender, C. S.: SNICAR-ADv3: a community tool for modeling spectral snow albedo, Geoscientific Model Development, 14, 7673–7704, https://doi.org/10.5194/gmd-14-7673-2021, 2021.

Hao, D., Bisht, G., He, C., Bair, E., Huang, H., Dang, C., Rittger, K., Gu, Y., Wang, H., Qian, Y., and Leung, L. R.: Improving snow albedo modeling in E3SM land model (version 2.0) and assessing its impacts on snow and surface fluxes over the Tibetan Plateau, Geoscientific Model Development Discussions, 1–31, https://doi.org/10.5194/gmd-2022-67, 2022.

He, C.: Modelling light-absorbing particle–snow–radiation interactions and impacts on snow albedo: fundamentals, recent advances and future directions, Environ. Chem., 19, 296–311, https://doi.org/10.1071/EN22013, 2022.

Onuma, Y., Takeuchi, N., Tanaka, S., Nagatsuka, N., Niwano, M., and Aoki, T.: Temporal changes in snow albedo, including the possible effects of red algal growth, in northwest Greenland, simulated with a physically based snow albedo model, The Cryosphere Discussions, 1–29, https://doi.org/10.5194/tc-2019-263, 2019.

Ryan, J. C., Smith, L. C., As, D. van, Cooley, S. W., Cooper, M. G., Pitcher, L. H., and Hubbard, A.: Greenland Ice Sheet surface melt amplified by snowline migration and bare ice exposure, Science Advances, 5, eaav3738, https://doi.org/10.1126/sciadv.aav3738, 2019.

Tedstone, A. J., Cook, J. M., Williamson, C. J., Hofer, S., McCutcheon, J., Irvine-Fynn, T., Gribbin, T., and Tranter, M.: Algal growth and weathering crust state drive variability in western Greenland Ice Sheet ice albedo, The Cryosphere, 14, 521–538, https://doi.org/10.5194/tc-14-521-2020, 2020.

Whicker, C. A., Flanner, M. G., Dang, C., Zender, C. S., Cook, J. M., and Gardner, A. S.: SNICAR-ADv4: a physically based radiative transfer model to represent the spectral albedo of glacier ice, The Cryosphere, 16, 1197–1220, https://doi.org/10.5194/tc-16-1197-2022, 2022.

Wientjes, I. G. M. and Oerlemans, J.: An explanation for the dark region in the western melt zone of the Greenland ice sheet, The Cryosphere, 4, 261–268, https://doi.org/10.5194/tc-4-261-2010, 2010.

Wientjes, I. G. M., Wal, R. S. W. V. D., Schwikowski, M., Zapf, A., Fahrni, S., and Wacker, L.: Carbonaceous particles reveal that Late Holocene dust causes the dark region in the western ablation zone of the Greenland ice sheet, Journal of Glaciology, 58, 787–794, https://doi.org/10.3189/2012JoG11J165, 2012.

Williamson, Cook, J., Tedstone, A., Yallop, M., McCutcheon, J., Poniecka, E., Campbell, D., Irvine-Fynn, T., McQuaid, J., Tranter, M., Perkins, R., and Anesio, A.: Algal photophysiology drives darkening and melt of the Greenland Ice Sheet, PNAS, https://doi.org/10.1073/pnas.1918412117, 2020.

---

## Author Comment (AC1)

RE: Referee # 1 comments on Light absorbing particles and snow aging feedback enhances albedo reduction on the Southwest Greenland ice sheet

Dear Raf Antwerpen and Editor,

The authors would like to thank the referee for the time and effort in reviewing and providing comments to our manuscript.

Below we include point-by-point response. For the comments the are related to technical text suggestions they will be included in the manuscript unless noted below. The reviewer comments are shown in cursive and purple highlight. Our response is in regular font.

**Response to Reviewer 1**

**General comments**

*Reviewer comment: I have noted some minor comments below, the most important of which pertain to the discussion section. I would be great to see a bit more discussion of the limitations of the measurements, the methods, and the conclusions you can draw from this analysis. It would for instance be good to describe what effects other LAPs (not included in the measurements, such as algae, brown carbon, and volcanic dust) could have on the LAP-snow aging feedback and the albedo reduction in general.*

Response: As suggested by the reviewer, we plan to include more discussion about the limations ofthe measurements, the methods, and the conlcusions in the revised manuscript. We may also include a section in the discussion about other LAPs and what potential effect they may have on snow aging and albedo.

*Reviewer comment: Consider writing in the present tense instead of the past tense. It could make the text a bit more active*

Response: As suggested, we will strive to rewrite the paper in present tense and active voice, except for the methods section which typically is better in past tense.

**Specific comments**

*L12: add algae to the list of LAPs.*
Done

*L15: explain briefly what the SNICAR model is.*
Done

*L20: do you mean enhance albedo reduction 4 to 10 times more?*
 Yes

*L31: elaborate a little bit more on "Greenland's summer albedo". Do you mean of the ice sheet in general ,of the snow, or of the ice?*

As requested, we will better explain that we mean the ice sheet in general when we write about Greenland's summer albedo

*L36: consider adding "global" before climate regulator.*
Done

*L39: is temperature a direct regulator of snow albedo or does it change albedo by changing the grain size/shape and water content?*
We will revised this text to better explain that temperture is an indirect regulator.n

*L43-44: remove "causing surface darkening". This aspect is covered in this sentence by "LAPs in snow reduce albedo"*
Done

*L51-L52: consider rewriting this sentence. It is not immediately clear to me what this means. Also make it clear that you are talking about CO2 and CH4 in the atmosphere, not in the ice/snow.*
As requested, we will rewrite this sentence to: The total climate forcing effect of BC, including accounting for all mechanisms that include direct, cloud and cryosphere effects is +1.1 W $m^{-2}$. In comparison, the total climate forcing effect ofCO$_2$ is +1.7 W $m^{-2}$ and CH$_4$ is 0.95 +W $m^{-2}$. The direct radiative forcing of BC in snow and ice has been estimated to be +0.13 W $m^{-2}$ (IPCC, 2019, Bond et al., 2013).

*L62: elaborate on why the 20-30 micrometer is important.*
Done

*L69-71: is it 16% of the average total mass loss (including dynamic losses)?*
This paper is only referencing to surface mass balance. We will rephrase to make this clear.

*L76: which LAPs did Lewis et al. consider?*
We will rephrase to better explain that Lewis et al. considered both BC and dust.

*L79-83: make sure the leap from "snow effective radius growth" to "snow aging" is clear. Right now it is not clear what "snow aging" is.*
We have rephrased this text to explain snow aging and better connect the text.

*L93-95: consider rewriting this sentence and elaborating a little bit on what SNICAR is and how you would use it to estimate this effect.*
We have rewritten this sentence to elaborate on wht SNICAR is and how we used it.

*L108: MAR is a regional climate model, not a reanalysis model.*
We have corrected this error

*L120: what happened at site G?*
We will rewrite the text to clarify that we were not able to collected duplicate samples here.

*L148-149: elaborate on what refractory BC is. Why can it be used as normal BC here?*
We will elaborate and add text to explain that "Normal" BC is typically used as a catch all term for different types of carbonaceous light absorbing particles, and that it refractory BC can be used as normal BC. Depending on the methodology used, BC is named elemental BC, equivalent BC, refractory BC or other terms. Elemental BC refers to an analysis making use of the thermally stable properties (up to 4000K in an inert atmosphere) of the carbonaceous

particles. Often aerosol mass or Raman spectroscopy are used in this method. The equivalent BC is referred to that mass concentrations derived from absorption coefficient measurements using a mass absorption coefficient constant. Lastly, rBC is derived from laser induced incandescence measuring the thermal emission of the carbon particle. The methodology used in this study is widely used and the measurments can be compared to recent measurements using other methods.

*L155: most of this is already mentioned before.*
Removed

*L157-158: what are the optimized operating parameters?*
We will clarify that the optimized parameters can be found in the cited references. Since there are quite a few parameters and standard values were used, we choose not to list them but refer to the references instead.

*L163: what does "adhering to outside of the detection range" mean? What are the particles adhering to?*
We are rephrasing this sentence to better explain that particle can aggregate to each other becoming larger and therefore becoming out of the particle size detection range (80-2000 nm)

*L180: is the 0.01mg the uncertainty?*
We are rephrasing to clarify that the 0.01 mg is the uncertanty.

*L188-189: the first part of the sentence is already mentioned before.*
As requested, we have rewritten this sentence.

*L195: mention the correlation coefficient in the main text.*
Done

*L203: what effects (due to LAPs) do you expect to see in the near infrared spectrum? Is it reasonable to not include this in your analysis? If so, show (with a reference) why.*
We have revised this sentence to clarify that we take into cconsideration NIR in the broadband albedo calculations.

*L206-215: 415-575 km is a long distance and the meteorology and climatology can be very different between these locations. How do you know the situation is comparable?*
We will rephrase this sentence to explain that Lewis's data is the best available to estimate optical grain size at our sites. Although Lewis's sites were 415-575 km away and may be influenced by other meteorological factors than our sites, they are at least from the same year as our observations.

*L234: consider merging figures 4 and 5. Right now figure 5 has almost all the information embedded in figure 4, except for the elevation which you could add in the caption.*
We agree with the reviewer that the figure 4 is redundant with figure 5. We have removed this figure and added the elevation information to figure 5.

*L242: is it winter snow grain shape or from May 2017?*
We will rewrite to clarify that althought the samples were collected in May, the snow was accumulated through the entire winter season from 2016 to 2017.

*L266: does BCtot relate to the BC concentration in a vertical slice in the snow pack, given that the unit is ng cm-2? If so, consider elaborating on that. It was not immediately clear to me.*

We will clarify this part to the reader to include that the total BC metric estimates BC deposition over a given area of the snowpack, independent of snow depth. Thus referring to a horizontal slice in the snowpack.

*L269: does average snow density refer to the density in each layer? If so, consider rewriting this sentence to make that clearer. It now seems like you take the average density of the entire layer.*
We will revised to clarify that average snow density is for the entire column.

*L301: consider rewriting "that snow ... albedo reductions are". It is not immediately clear to me what you mean here.*
We will revise to clarify that this refer to the albedo reductions for BC, dust, and LAP considering snow metamorphism

*L350: elaborate on the BC concentrations in southeast and southwest Greenland. These regions are very different and might not be easily comparable. Given what you know about the sources and transport routes of BC to Greenland, elaborate on what differences/similarities would you expect to find between BC concentrations in southeast Greenland vs what you found in southwest Greenland? Take into account the general environment, elevation and topography of the sample locations in southeast and southwest Greenland. e.g. what side of the ridge of the GrIS each location is on and what is the general wind pattern from source to deposition location. Could local sources be affecting the BC concentrations (such as for dust at Dye-2)?*
As requested, we will elaborate more on the BC concentrations and the difference between SE and SW Greenland ice sheet. We will explain that larger precipitation and general wind patters could lead to more dust and BC depositions in SE Greenland. We will also better explain that local sources could be affecting dust concentrations at Dye-2.

*L405: briefly repeat the results of Lewis et al. 2021 here.*
As requested, we will briefly repeat Lewis et al., 2021 results here

*L413-415: move this to results. Also explain how you get the value 1.18.*
As requested, it has been moved to results and we will also better explain how we got the 1.18 factor. This factor was obtained from the difference in albedo reductions between SA and non SA scenarios at the maximum albedo reduction wavelength (at $\lambda = 440$ nm (Table 3).

*L417-419: This statement is not immediately clear to me. If half of the measurements were under clear-sky conditions (and, thus, the other half under (partly) overcast conditions), would this not mean that your results do not solely apply to clear-sky conditions?*
We will clarify this statment to clarify that while some of our sampling occurred during partly overcast conditions, the SNICAR model was run under clear-sky conditions as the SNICAR-ADv3: Online Snow Albedo Simulator only differentiates between direct or diffused incident radiation without any option for partial overcast conditions.

*L421-422: what do you mean exactly with this? Are you talking about interannual variability of albedo reduction? If so, that cannot lead to greater albedo reduction over summer, it would the other way around. Greater albedo reduction in summer leads to interannual variability of albedo reduction. Or do you mean interannual variability of another variable? If so, make clear what you are referring to here.*
As requested we will rewrite to clariify that greater albedo reduction in summer leads to internannual variability of albedo reduction.

*L423: elaborate what you mean with snow melting processes as a factor leading to albedo variability.*

As requested, we will elaborate to explain that we here efer to snow melting ability to darkern surfaces by through enlarged grain size, water pools, and conversion of snow to ice.

*L425: does the 3.0 ± 0.4% refer to the difference between summer and spring albedo reduction? Consider mentioning both average albedo reductions for spring and summer to show the difference.*
We will revise the text to clarify that we here mean to the difference between impurity laden snow and clean snow under the same conditions, we will also mention average albedo reductions for both here.

*L429: what is the water year?*
We will explain that the water year is defined as the period between October 1st to September 30th

We will also incorporate all the minor technical and editorial suggestions by the reviewer in the revised manuscript, but we will not list all of them here.

---

## Author Comment (AC3)

RE: Referee # 2 comments on Light absorbing particles and snow aging feedback enhances albedo reduction on the Southwest Greenland ice sheet

Dear Referee and Editor,

The authors would like to thank the referee for the time and effort in reviewing and providing comments to our manuscript.

Below we include point-by-point response. For the comments the are related to technical text suggestions they will be included in the manuscript unless noted below. The reviewer comments are shown in cursive and purple highlight. Our response is in regular font.

**Response to Referee 2**

**General Comments**

*Use the multi-layer SNICAR-ADv3 or 4 (both are available online). While properties lower in the snow pack will only minimally influence the albedo, it would be beneficial to utilize the in situ measurements with depth. It would also be nice to see the modeled output albedo for the simulations constrained by measurements, with the caveat that the true albedo is not known.*
We will clarify that we used SNICSR-ADv3 and that the simulations were constained by field measurements. As properties low in snowpack do not significantly influence albedo we did not included them here.

*Address the influence of grain size with regard to the results presented in figure 7, the manuscript states that both grain size and grain shape change with snow metamorphism and both have implications on albedo. However, only snow grain shape is addressed with the measurements taken. It is also important to provide justification of why hexagonal grains are used to represent fresh snow and spheroid grains are used to represent aged snow.*
We will revise the manuscript to further explain the selection of snow grain shapes (hexagonal and spheroid grains to represent fresh and aged snow, respectively) is rooted in the fact that the most basic and known variety of fresh snow is the simple prism, displaying hexagonal symmetry. However, as it ages snow grains become more rounded and hence we chose spheroid grains. Although we dont try to estimate the influence of grain size, given that we our field measurements only have optical grain size during measurement, we can include some context to the reader better understand the influence of grain growth with regard to the results presented in figure 7 and connect it to our sensitivity study presented in Figure 8.

*It is unclear what the relationship between LAP and snow metamorphism is, for example, what concentration of LAP justifies a transition from hexagonal shaped grains to spheroidal grains? How much of the grain growth and shape change can we attribute to LAPs rather than environmental variables such as temperature?*
We will clarify that this the aim of this study is not to quantify the grain growth due to environmental variables such as temperatures vs LAP. Instead, the goal of this study is to study LAP influence over the naturally occurring snow metamorphism process. Our findings show that considering LAP in snow metamorphism process enhances impact on albedo reductions.

*Please provide a more thorough justification for the snow properties and LAP concentrations used to simulate the results presented in figure 8. These results are all theoretical as there is no justification of the snow conditions used to simulate the albedo differences.*
We will clarify that the Figure 8 values are theoretical, and that the purpose is to provide a sensitivity test of the properties to a range of concentrations under differing snow properties. We will rewrite the manuscript and explain

that we selected the range of observed snow properties in previous campaigns and literature (i.e reff as little as 50 to and as large as 550) and theoretical snow densities for various stages of snow metamorphism. (Muskett, 2012)

*There needs to be significant clarifications within the methods for how the change in albedo is calculated. It is unclear how the total changes and % changes in albedo are calculated, and as a result these results are not easily reproducible.*
We will revise the manuscript to clarify our method to calculate % changes in albedo, i.e. the albedo reduction was calculated as the difference of broadband albedo values between clean snow and snow containing impurities (BC+dust) as a fraction of the two combined.We will reinforce this when the results are presented.

*I think the introduction could include more discussion about other measurements and how the measurements described here are different and useful (the measurements were taken with depth, you measure the grain size, shape, and density of the snow).*
As suggested, we have added discussions about this in the revised manuscript.

**Specific comments:**

*Line 19-20: "SNICAR simulations constrained by our measurements show that LAP-snow aging feedback reduce albedo reduction 4 to 10 times more than previously thought," Please add more nuance than "more than previously thought". For example, are you comparing to a study that also used measurements and RT modeling? Then say "more than previous studies have estimated" It's also important to elaborate on how this range was calculated in your methods / results.*
As suggested, we will add more context here.

*Line 33-34: "However, sparse measurements of LAPs from the Greenland ice sheet snow limits our understanding of the LAP and Greenland albedo reductions" it could be useful to briefly discuss past measurement campaigns and different LAP data and to move your discussion of previous measurements to the introduction and then in the results you can say your measurements compared well and explain the possible discrepancies (Bøggild et al., 2010; Wientjes et al., 2012; Wientjes and Oerlemans, 2010; Cook et al., 2020; Onuma et al., 2019).*
As suggested, we will make sure to include these references

*Line 53-55: "BC and other carbonaceous aerosols could reduce projected temperature increase by 0.5°C by 2050 while preventing millions of premature air pollution-related deaths and crop losses. (Samset et al., 2014; Lee et al., 2013; UNEP & WMO, 2011)." The removal of BC?*
As suggested, we will clarify this sentence to explain that it is the removal of BC that could result in these beneficial impacts.

*Lines 61-64: "During transport particles may fractionate and reduce the size of the dust particles transported over long distances to less than 20 to 30 μm (van der Does et al., 2018; Ryder et al., 2013; Kok et al., 2012). Evidence suggests that Arctic dust comes from Gobi and Taklaman deserts in Asia (38%), Saharan dust in Africa (32%), and local high latitude sources (27%) (Takemura et al., 2009)." This doesn't seem very relevant. It would be more useful to provide background on relevant measurements in this region of the ice sheet.*
We agree with the reviewer and will remove this part from the manuscript and provide additional measurements in this region where available.

*Snow and ice algae and the exposure of bare ice/crustal surfaces have been found to be very strong albedo reducers, I think your discussion should include some mention of these other mechanisms (Chevrollier et al., 2022; Tedstone et al., 2020; Ryan et al., 2019)*
As suggested, we will write about the suggested other mechanisms in the discussion

*Line 83-86: "While global climate models include snow radiative transfer computation, including parametrizations of snow physical characteristics (particle size, particle shape, impurity load and solar zenith), the positive feedback of LAP on snow is often unaccounted in Greenland Ice Sheet measurements (Saito et al., 2019; He et al., 2018; Yasunari et al. 2011; Gardner and Sharp 2010; Marshall and Oglesby 1994)." He and others have done significant work to improve this representation, it is likely worth mentioning their work (He, 2022; Hao et al., 2022). It is also unclear if this sentence is about the importance of this representation in model or measurements. Please adjust.*

As suggested, we will incorporate the work by He, 2022 and Hao el al. 2022 and rewrite so that it is clear that this text is about modelling.

*Line 93: "Snow, Ice, and Aerosol Radiative (SNICAR) model (Flanner et al., 2007)" It is unclear which version of SNICAR is used in this study. The 2007 version does not allow for various grain shapes. There are multiple newer versions of this model that have been recently released that allow for multi-layer simulations and various grain shapes (Whicker et al., 2022; Flanner et al., 2021; Dang et al., 2019). These newer versions do not utilize the Toon 1989 solving method. Please clarify in the methods.*

We will revised the manuscript to clarify that SNICSR-ADv3 was used in this study

*Line 200: "We used the single-layer Snow, Ice, and Aerosol Radiation (SNICAR) model to estimate snow albedo" It would be more interesting if you used a multilayer approach using SNICAR-ADv3 or SNICAR-ADv4 and utilized your snow measurements with depth*

Althought a multilayer approach and understanding the snow measurements with depth is an interesting question, we choose to focus on single layer applications in the study. We will add text in the discussion to propose using a mulitlayer methods as an extension of this study.

*Line 216: "These values of reff are used in the SNICAR model to bracket the likely range of actual reff." There was no use of a range of reff in the results? If you were going to use 146.2 ± 28.8 as a range you would need to run 3 simulations, one for reff of 146.2-28.8, one for 146.2, and one for 146.2+28.8. This is also different from the values reported in table 2*

We will clarify in the manuscript that we ran simulations with many values between each range and that model results are reported as average. We will also correct the value to 146.7.

*Figure 5 is great, it could be interesting to try and overlay [dust/BC] so we can see all the snow properties in one figure, or at least improve the dust/BC figure to look similar to figure 5.*

We will be keeping Figure 5 and 6 separate to avoid cluttering the image with too much information but will work on Figure 6 to look similar to Figure 5.

*Figure 7 caption: Please find a way to more clearly represent what delta albedo you are showing. It is unclear which figures include changing snow grain shape and which only include the influence of LAPs. For example use a more descriptive ledged like "spheroid grains w/ LAPs – hexagonal plate w/ LAPs". The last line of the caption makes it seem like the same grain size and shape was used for all simulations "All spectral albedo changes represent the difference between the LAP impacted snow and the clean snow simulated with the model using the same snow properties." It is also unclear which snow properties are used in 7d, please refer to the corresponding table.*

As requested, we will provide a better explanation in the figure caption.

*Table 2: The same grain size was used for all (both SA and non-SA) simulations? If that is the case, these simulations don't capture the full influence of snow metamorphism. Why is the average dust concentration used but only the surface BC used in the simulation? Also the effective radius 146.7 is not an option within SNICAR – please adjust. Please also include all SNICAR parameters so these results can be recreated. For example, make a SA and non-SA column with all model input parameters (including the varying LAPs) so these simulations can easily be*

*recreated and the differences between the simulations are clear. Please also apply this technique to the SNICAR parameters used for the figure 8 simulations.*

Regarding Table 2, we agree that using the same grain size for all simulations (both SA and non-SA) may not fully capture the influence of snow metamorphism. We acknowledge that this is a limitation of our study and will include a note in the manuscript to this effect. To this effect we performed the sensitivity study (which is presented in Figure 8) to evaluate the impact of different grain sizes and LAP content in albedo reductions. We will make this more clear in our final manuscript. Regarding the use of dust and surface BC in the simulations, we only used surface BC because it tends to agglomerate in the surface due to its hydroscopicity. On the other hand, dust can be eluted by percolation, so we used the average dust concentration to accurately represent the water year period. We will clarify this in the manuscript. We apologize for the error in reporting the effective radius in SNICAR. We will correct this to 147, which is what SNICAR automatically converts 146.7 to. We will also include all SNICAR parameters in the manuscript so that our results can be easily recreated. Specifically, we will add a SA and non-SA column with all model input parameters presented in Table 2 and 4 so that the our snow aging treatment and difference between the simulations is clear. We will also create a table for the SNICAR parameters used in Figure 8 simulations.

*Line 300-303: "We observe that snow aging related BC, dust, and LAP combined albedo reductions are 2.6±0.5, 1.18±0.06, and 1.18±0.04, respectively, times greater than those related to non-SA simulations." These albedo reductions are simulated, not observed. It is also not clear which simulations these reductions are based on. Why is the BC only reduction larger than that of the LAP combined albedo reduction?*

We agree that using the term "observed" might not be appropriate and we will change it to ensure is clear albedo reductions are simulated. The clarifications for Table 2 explained in the previous comment might help clarify to the reader the nature of this observations. Regarding the relative magnitudes of the albedo reductions, it is not uncommon to find higher albedo reductions by the BC component than the combined component because given the non-linearity relationship, as light penetration increases with LAP concentration (Dang et al., 2017).

*Line 320: "Using previous studies' summer measured snow grain size of 550 µm (Warren, personal correspondence) and the LAP concentrations in the depth hoar layer, detailed in Table 4, we parameterize the SNICAR model to estimate the spectral albedo reductions for summer 2016 (Figure 7d) " Table 4, The density values in table 4 are not representative of the summer 2016 values as the snowpack has been compacted by melting, refreezing, and fresh snowfall. While the density doesn't influence the albedo of snow in SNICAR, it is likely worth mentioning that this density is greater than that of the 2017 snow density and making sure the reader understands that this will not have an impact on the delta albedo calculations. Please explain why using the snow grain radius from Warren is better than following the method used for the 2016 grain size, the 550um grain size is much larger than the 146.2 um grain size estimate.*

We will revised the paper and explain that we selected Warren's snow grain radius because these measurements were done in the Summer, which is our target season. In the revised paper, we will better explain that we use the bottom layer concentration because it is representative of summer 2016 conditions. The grain size observed in the bottom layer of our study may not be representative of the time when this layer (2016) was on the surface, as it could have been impacted by compaction and post-depositional processes. Therefore we used Warren 550um grain size as it is representative of summer melting snow conditions and the impact LAP can have in that season.

*Line 336: "f) Broadband albedo reductions for snow with LAP at reff of 50,110,200,350,and550µm." This simulation only includes BC, correct? Please adjust the legend accordingly.*

We will adjust the legend and better explain that the simulations includes BC and dust, and that the concentrations used for only-dust simulations are in parenthesis.

*Line 331-332: "Upper boundary (dotted lines) correspond to modelled albedo reduction assuming BC(dust)coated and lower boundary values correspond BC (dust) uncoated" this seems like you're getting the same albedo reduction for BC and dust? As you're using one line to represent different concentrations of both. Are you using both*

*BC and dust in one simulation? I think your units for dust concentration are incorrect on the figure. Are you using milligram/gram (mg/g) or microgram/gram (ug/g)?*

We will revise to clarify that we were using both concentrations in one simulation. We are using ug/g (micrograms/gram)

*332-334: "a)reff 50μm,60kgm-3 (fresh snow); b) 110 μm, 150 kg m-3 (slightly aged); c) 200 μm, 250 kg m-3 (settled snow); d) 350 μm, 375 kg m-3 (wind packed snow); e) 550 μm, 600 kg m-3 (melting snow)." There is no justification for why these specific snow properties are chosen for aged/settled/wind packed/melted snow. Please include citations and discussion for why these snow properties were chosen for each snow state. Please also justify the concentrations of dust and BC you are using. In your measurements, I see a max dust concentration of ~3ug/g and a max BC concentration of ~2ng/g, but in your figure 8 simulations you are using up to 200ug/g (I am assuming mg/g on the figure is a typo) of dust and 1500 ng/g of BC. The use of such high LAP concentration inflates the difference in albedo and these concentrations seem unjustified based on the measurements presented here and your comparison to previous measurements.*

As suggested, we will revise the manuscript to include citations and explain that all values for Figure 8 were selected to provide a sensitivity study and these are theoretical values not obtained from literature.

*Line 334:335: "This simulations are computed as the albedo of pure snow (hexagonal shape) minus the difference of the albedo of pure snow (spheroid shape) the albedo of LAP- containing snow (spheroid shape)." It is unclear what differences are being represented here, it might be more clear if you also show the SNICAR output albedo with sufficient legends to be more clear about which albedos you are differencing and/or write out an equation. It sounds like you are differencing three albedo simulations, which does not seem correct.*

As suggested, we will clarify this sentence as it should read his simulations are computed as the albedo of pure snow (hexagonal shape) minus the albedo of LAP-containing snow (spheroid shape)

*Line 340: "LAPs have double the impact on albedo reductions compared to fresh snow (Figure 8)." Please include quantitative results and range, this seems like it is only true for very large LAP concentrations? If so, this result is contingent on the justification of both snow properties and LAP concentrations used in figure 8.*

We will rewrite this sentence and provide quantitative results and range. This sentence was specifically referring to Figure 8e but we agree it could be improved for clarification.

*Line 345: "Thus, compared with fresh snow, BC concentration of 0.5 ug g-1, and dust concentration of 1.0 ug g-1 causes an additional net albedo reduction of 0.01 - 0.15 depending on the impurity content in melting snow under summer conditions." Why does this depend on impurity content? Isn't this result for a simulation with BC concentration of 0.5 ug g-1, and dust concentration of 1.0 ug g-1? Are you comparing fresh snow with LAP to aged snow with LAP?*

Indeed this sentence is poorly worded. We will revise to clarify that we are comparing fresh snow with out LAPs with aged snow with LAP.

*Line 398-399: "While our measured LAP concentrations were relatively low, we find larger albedo reduction than previous studies. " Do you have citations for these previous studies? I'm not sure this is fair to say because no in situ albedo measurements were taken. It is important to note alongside this discussion of the impact on albedo that the uncertainty associated with this albedo is unknown because no in situ albedo measurements were taken. Also, are theses albedo reductions based on theoretical snow properties and LAP concentrations (ie not measured values in figure 8)? If so that major caveat needs to also be mentioned.*

We will revise the text to clarify that our albedo estiamtes are made with a model, while previous studies are based observations. We will also add the citations. We will also clarify that the the albedo reductions referred to here are estimated with model simulations using measured snow properties.

*Line 403: "while the isolated snow grain shape change (without LAP) influences mostly the albedo at the shorter wavelengths (Figure 7d)" What wavelengths and by how much? Changes to grain shape and size have the strongest impact in the NIR, so seems counterintuitive to not include this region of the spectra in your analysis. Figure 7d does not show the impact of changing snow grain shape.*

We will revise the text to clarify that a section of NIR is included in figure 7 (i.e. 700 - 1000 nm). We will also add text to explained that grain shape strongly influence NIR and cannot alone explain the difference in the VIS region.

*Line 404-405: "Therefore, after subtracting the clean snow grain shape change effect we can assume that all the observed absorption is due to the presence and snow metamorphism influence of LAP" This absorption is not observed, it was simulated. It is also unclear what you are trying to say here*

As requested, we will clarify this sentence.

*Lines 405-407: "Contrary to the findings of Lewis et al. 2021, we found that LAP have a significant role in the albedo reduction considering their role on accelerating snow metamorphism which amplifies LAP radiation perturbation (Schneider & Flanner, 2017; Hadley & Kirchstetter 2012)." I'm not sure this is fair to say, the role LAP's play in accelerating snow metamorphism is not clear from this study. You do not calculate how the LAPs are influencing the grain shape, you simulate the impact the combined effect of LAPs and snow metamorphism have on albedo. Can you suggest the rate of metamorphism based on only LAPs (not the influence of temperature or general snow aging)?*

We disagree with the reviewer. The aim of this study is not to quantify the influence of LAP on grain shape, but show that LAPs influence grows as snow ages. As can be seen in figure 7,  snow aging by itself cant explain the albedo reduction over the VIS region. We will work on the discussion section to make this argument more clear.

*Line 417-419: "Our results only apply for direct incident radiation given that half of our measurements were under clear-sky conditions. We note that overcast conditions in Greenland would render a slightly lower albedo perturbation than our results indicate." Why is this relevant? Wouldn't the sky conditions only be relevant for albedo measurements?*

We will reword this part to explained that it is relevant to mention the sky conditions because it is a parameter of the SNICAR albedo simulations.

*Lines 432:434: "Using our observed BC and dust concentrations values and a model that considers snow metamorphism, we found that albedo is reduced by as much as 3.0 ± 0.4% in the summer and by 1.9 ± 0.1% as an annual average in contrast to fresh snow without LAP" It seems as though in the simulations where measured dust and BC are used (figure 7) only changing snow grain shape is accounted for, there is no grain growth, so it doesn't seem accurate to say "the model considers snow metamorphism".*

We agree this model does not fully capture snow metamorphism. The reasoning behind our choice is that we don't have an accurate way to determine the size of fresh snow without impurities or estimate  the grain growth rate. The inclusion of snow grain size growth would very likely result in even greater albedo reductions, thus, we wanted to avoid non-observation estimations that could lead to overestimations given that snow grain size plays such a significant role on snow albedo. We will make  sure that this limitation and its impact on the interpretation of results is much clearer to the reader on the discussion.

*Lines 434-436: "These albedo reductions are 4 to 10 times larger than previous studies with similar LAP concentrations and suggests that LAPs have a greater impact on surface melting through snow metamorphism than previously thought" Please explain how the 4 to 10x is calculated.*

We will revised the manucript and explain in more detail how the calculations were done

All other minor technical and editorial suggestions will be corrected in the revised manuscript.